# Go4RL: Improving the Pre-training Data Mixture of Large Language Models for Enhancing Reinforcement Learning

## Abstract

The development of reinforcement learning (RL)-trained large language models (LLMs) such as OpenAI-o1 and DeepSeek-R1 has demonstrated notable advancement in solving complex logical tasks involving mathematics and programming. Existing studies, however, show that the reasoning capability of RL-trained LLMs varies across base model families, raising the critical research question: what base model is suitable for augmenting RL's reasoning capabilities? Recent prevailing research shows that 1) the reasoning capacity of the RL-trained model remains bounded by that of its base model and 2) the data mixture used for pre-training has a significant impact on the base model's performance. Building on these insights, we propose Go4RL, which seeks to answer the above question by investigating how pre-training dataset mixture strategies relate to a viable base model for RL training. Go4RL first defines the measurement of base models' reasoning capability boundaries as the average perplexity scores of the base LLM on the RL-generated $k$ responses, denoted by avg(k-ppl). Then we formulate finding the optimal pre-training dataset mixing ratios for RL as a regression task, using the proposed avg(k-ppl) as the fitting objective instead of the traditional language modeling loss. Finally, we use the learned regression model to predict the performance of unseen mixtures and apply the best predicted mixture to train a large-scale model that achieves better RL performance. We train both pre-trained and RL-trained proxy models with 1M parameters for regression fitting and then scale up to 1B parameters using various data mixtures to validate Go4RL. The experimental results on both online and offline RL algorithms show that the optimized data mixture predicted by Go4RL yields a better base model for RL training.

## 1 Introduction

The recently released large language models (LLMs), including OpenAI-o1 (Jaech et al., 2024) and DeepSeek-R1 (Guo et al., 2025), have demonstrated superior skill in accomplishing intricate reasoning tasks, particularly in mathematics and programming. The key to these advancements is reinforcement learning (RL) with verified rewards or human preferences, which has emerged as a pivotal phase in the post-training process of the base LLM for establishing enhanced reasoning abilities (Dang et al., 2025; Shah et al., 2025).

Despite the remarkable capabilities of RL-trained LLMs, the underlying mechanisms driving their enhancement remain unclear and motivate active debate among the research community. Some studies (Liu et al., 2025; Yue et al., 2025; Shah et al., 2025) examine whether reinforcement learning creates new reasoning solutions from scratch or merely applies existing patterns in base models. Others (Wang et al., 2025; Zhao et al., 2025) demonstrate that the reasoning capability of RL-trained LLMs varies among base model families, motivating more investigation into which base model is best suited for boosting RL's reasoning capabilities. While prior research has not reached a consensus on whether reinforcement learning truly expands the reasoning boundaries of base LLMs, these studies acknowledge that the efficacy of RL is significantly dependent on the initial capabilities of the base model.

Building on the above research consensus, this work seeks to further explore what base model is suitable for RL from the perspective of pre-training dataset mixture strategies. Our focus on the pre-training data mixture stems from prior studies demonstrating its significant impact on base model capabilities (Liu et al., 2024; Xie et al., 2023; Ye et al., 2024b) and subsequently on RL performance (Wang et al., 2025; Zhao et al., 2025). Another issue we need to address is how to choose a suitable base model for RL. There are no straightforward evaluation metrics, such as loss functions, due to the two phases of pre-training and RL training, as well as the complex and unstable RL process (Rafailov et al., 2023; Shao et al., 2024). Inspired by the pass@k metrics, which are prevailing adopted for evaluating RL-trained LLMs (Liu et al., 2025), we propose to utilize the average perplexity of the base LLM on the RL-generated $k$ responses to evaluate reasoning boundaries of the base model. Such evaluation is agnostic to the specific RL algorithm utilized and may be adapted to any variety (e.g., offline algorithm GPO and online algorithm GRPO), enhancing its versatility.

In summary, we propose an RL algorithm-agnostic method, Go4RL, which first defines the measurement of base models' reasoning capability boundaries as the average perplexity scores of the base LLM on the RL-generated $k$ responses, denoted by avg(k-ppl). Then Go4RL formulates finding the optimal pre-training dataset mixing ratios for downstream RL as a regression task (Liu et al., 2024), using the proposed avg(k-ppl) as the fitting objective. Finally, we use the fitted regression model to predict the performance of unseen mixtures and apply the best predicted mixture to train a large-scale model that achieves better RL performance.

To validate Go4RL, we follow Liu et al. (2024) to train proxy models (both pre-trained and RL-trained) with 1M parameters and scale up to 1B parameters using different data mixtures from three domains (general corpus, math, and coding). Each base model is fine-tuned using RL algorithms directly without supervised fine-tuning. The experimental results on both representative online (GRPO (Shao et al., 2024)) and offline (DPO (Rafailov et al., 2023)) RL algorithms show that the optimized data mixture predicted by Go4RL yields a better base model for RL training.

## 2 RELATED WORKS

**The relationship between the RL and base model**. The success of RL-trained LLMs inspires research interest in exploring their underlying mechanism. Existing studies (Yue et al., 2025; Dang et al., 2025; Zhao et al., 2025; Liu et al., 2025) investigate whether reinforcement learning creates new reasoning solutions from scratch or merely applies existing patterns in base models. They suggest that RL-trained models do not gain new reasoning skills beyond those seen in their base models but rather enhance sampling efficiency along right reasoning paths. On the contrary, ProRL (Liu et al., 2025) offers surprising new insights that RL can indeed discover genuinely new solution pathways entirely absent in base models when given sufficient training time and applied to novel reasoning tasks. While the prior research has not reached a consensus on whether RL truly expands the reasoning boundaries of base LLMs, they acknowledge that the efficacy of RL is significantly dependent on the initial capabilities of the base model. Wang et al. (2025) demonstrates that the reasoning capability of RL-trained LLMs varies among base model families, such as Llama (Dubey et al., 2024) and Qwen (Yang et al., 2025). It further proves that the base LLMs could be enhanced for RL through mid-training with a proper data mixture. Differently, we explore how to get an RL-friendly base LLM during the pre-training. Zhao et al. (2025) seeks to clarify how the composition of pretraining data affects the efficacy of RL fine-tuning using varying proportions of the text and math datasets for pre-training. They focus on analyzing the RL reaction on the pre-training data instead of exploring an optimal pre-training data mixture.

**Data mixing.** Existing LLMs require a huge amount of training data from different data sources to achieve remarkable capabilities. It has been proved that the data mixing strategies have a great impact on the LLMs' performance (Albalak et al., 2023; 2024). As the field progressed, learnable task-performance-driven mixing methods have gradually gained popularity (Schioppa et al., 2023). Task-specific mixing methods generally fall into two categories: offline approaches determine domain weights through proxy model evaluation (Ye et al., 2024b; Liu et al., 2024), while online methods dynamically adjust weights based on feedback during the target model's training (Chen et al., 2023; Xie et al., 2023; Fan et al., 2023). Notably, recent works have leveraged scaling laws or regression models to provide guidance for forecasting data mixing performance in large-scale models (Hoffmann et al., 2022; Muennighoff et al., 2023; Liu et al., 2024). We choose the regression

model (Liu et al., 2024) over the scaling law (Ye et al., 2024a) to forecast appropriate data mixture ratios for large-scale LLMs due to a lack of straightforward evaluation metrics for our problem. Our work has a different fitting target with Liu et al. (2024).

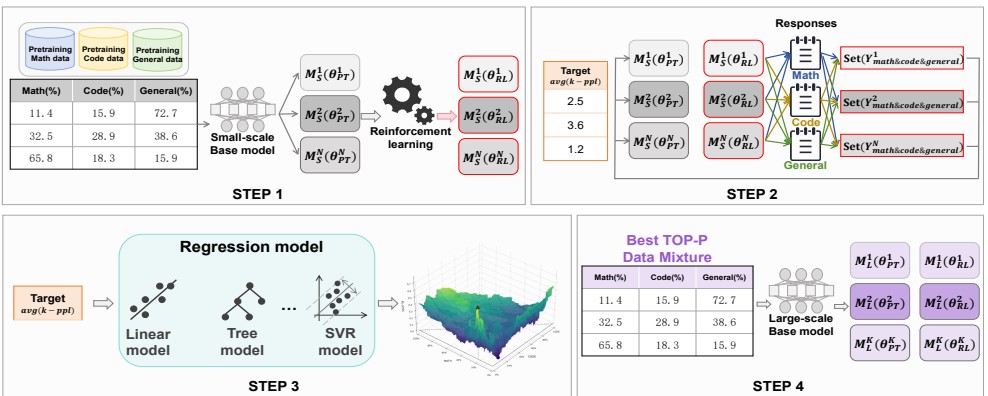

Figure 1: The framework of Go4RL, including the following steps: 1)Generate random pre-training data mixtures of math, code, and general domains, then pre-train a series of small-scale proxy models based on these mixtures and continually train these proxy models through reinforcement learning. 2) Obtain the target average perplexity avg(k-ppl) of the pre-trained model over $k$ responses generated by the RL-trained model. 3)Fit a regression model using the ratios of mixtures and the avg(k-ppl) as the target label. 4)Leverage the fitted regression model to predict the best mixture for the target value on a large-scale model and further validate the prediction.

## 3 METHODOLOGY

### 3.1 OVERVIEW

The scenario in this paper is to identify the optimal pre-training data mixture ratio that yields the best-performing LLM after both pre-training and reinforcement learning stages. As illustrated in Figure 1, the proposed Go4RL involves the following four key steps. (1) We first train small proxy models, which will be used later for regression. These small proxy models include both pre-trained and RL-trained models. The pre-trained models are trained using numerous different data mixtures from three domains (general corpus, code, and math). It should be noted that our proposed framework is easy to scale to more domains. The RL-trained models are directly trained from the pre-trained ones. These settings follow the SOTA reasoning model DeepSeek R1 (Guo et al., 2025), which applied RL directly to the base pre-trained LLMs, bypassing any intermediate supervised tuning. The data mixture domains are also predominantly adopted by open-source LLM families such as Llama (Dubey et al., 2024) and QWen (Yang et al., 2025). (2) With the small proxy pre-trained and RL-trained models, we calculate the average perplexity scores of pre-trained models on $k$ responses generated by RL-trained models, regarded as avg(k-ppl). (3) Fit a regression model using the data mixture ratios as features and the avg(k-ppl) calculated in step 2 as the target label of the regression. (4) We use the fitted regression model to predict the optimal data mixture on large-scale LLMs and then train models to validate. Next, we will elaborate on the specific details of each step.

### 3.2 TRAIN SMALL-SCALE PROXY MODELS WITH RANDOM DATA MIXTURES

First, we adopt numerous diverse data mixtures to obtain a series of small-scale proxy models, including both the pre-trained base models and RL-trained models. The mixing ratios are sampled based on a token-distribution-based Dirichlet distribution, ensuring broad coverage of weight extremes while preventing overemphasis on low-frequency token domains.

Specifically, leveraging the Dirichlet distribution (Minka, 2000), we randomly generate $N$ different data mixture ratios across three training domains (math, code, and general) to construct a wide

spectrum of sparse and near-uniform distributions:

$$Set(R) = [R^1, R^2, ..., R^n, ..., R^N]$$
$$R^n = [r^n_{general} : r^n_{code} : r^n_{math}] \tag{1}$$

$R^n \in Set(R)$ denotes the possible pre-training data mixture ratio of three domains (e.g., $R^n = [6 : 2 : 2]$).

A set of $N$ small-scale proxy pre-trained models $Set(M_S(\theta_{PT}))$ could be obtained, with each $M_S$ pre-trained by adopting each data mixture strategy $R^n \in Set(R)$:

$$Set(M_S(\theta_{PT})) = [M^1_S(\theta^1_{PT}), M^2_S(\theta^2_{PT}), ..., M^N_S(\theta^N_{PT})] \tag{2}$$

With $Set(M_S(\theta_{PT}))$, we then directly apply reinforcement learning with the same setting to train each pre-trained proxy model, ultimately obtaining a series of RL-optimized small proxy models:

$$Set(M_S(\theta_{RL})) = [M^1_S(\theta^1_{RL}), M^2_S(\theta^2_{RL}), ..., M^N_S(\theta^N_{RL})] \tag{3}$$

### 3.3 CALCULATE THE AVERAGE PERPLEXITY AVG (K-PPL)

The pipeline for RL-trained LLMs consists of two phases: pre-training and RL training. Furthermore, RL is a sophisticated and unstable approach that has multiple variations, including both online and offline algorithms with distinct techniques such as DPO (Rafailov et al., 2023) and GRPO (Shao et al., 2024). As a result, there are no straightforward evaluation metrics for analyzing the data mixture effect on RL-trained models via base models, like the predict-next-token loss function used to evaluate base LLM data mixing.

Inspired by the pass@k metrics, which are widely used to evaluate RL-trained LLMs (Guo et al., 2025; Liu et al., 2025; Zhao et al., 2025), we propose using the average perplexity of the base LLM on the RL-generated $k$ responses to assess the base models' reasoning ability boundaries, named avg(k-ppl). Such avg(k-ppl) evaluation is agnostic to the specific RL algorithm utilized and may be adapted to any variety (e.g., offline algorithm DPO and online algorithm GRPO), enhancing its versatility.

Generally, given a base proxy model $M^n_S(\theta^n_{PT})$, its corresponding RL-trained proxy model $M^n_S(\theta^n_{RL})$, an input $x$, and a response $Y = (y_1, y_2, ..., y_T)$ that is generated by the $M^n_S(\theta^n_{RL})$, the perplexity of the pre-trained model for $Y$ is defined as the exponentiated average negative log-likelihood of a sequence:

$$PPL_{M^n_S(\theta^n_{PT})}(Y|x)$$
$$= exp(-\frac{1}{T}\sum_{t=1}^{T} logP(y_T|(x, y_1, ..., y_{T-1}))) \tag{4}$$

Eq.4 reflects the model's ability to predict the given response $Y$ conditioned on the prompt $x$. Lower perplexity indicates that the model has a higher likelihood of generating this response.

We randomly sample a set of prompts in three domains: $x \in X = [Set(X_{code}), Set(X_{math}), Set(X_{general})]$. Similar to the pass@k (Liu et al., 2025) metrics for evaluating RL performance, we use each $M^n_S(\theta^n_{RL})$ to generate $k$ multi-responses for each prompt $x$. The set of all these responses is denoted as $Set^n_{RL}(Y)$. Let $\overline{k-ppl}^n_{PT}(Y_{RL})$ be $M^n_S(\theta^n_{PT})$'s average perplexity of these responses generated by $M^n_S(\theta^n_{RL})$, defined as below:

$$\overline{k-ppl}^n_{PT}(Y_{RL}) = \frac{\sum_{Y_\alpha \in Set^n_{RL}(Y)} PPL_{M^n_S(\theta^n_{PT})}(Y_\alpha|x_\alpha)}{l_1 + l_2 + l_3}$$
$$PPL_{M_S} = [\overline{k-ppl}^1_{PT}(Y_{RL}), ..., \overline{k-ppl}^N_{PT}(Y_{RL})] \tag{5}$$
$$Set^n_{RL}(Y) = Set(Y^{l_1}_{code}) \cup Set(Y^{l_2}_{math}) \cup Set(Y^{l_3}_{general})$$

Where $l_1, l_2, l_3$ respectively represent the number of $M_S^n(\theta_{RL}^n)$'s responses in three domains. $Y_\alpha$ denotes each response in $Set_{RL}^n(Y)$, while $x_\alpha$ is its corresponding prompt.

The definition of avg(k-ppl) can effectively evaluate the reasoning capabilities boundaries of pre-trained models (Liu et al., 2025). Once we have obtained these avg(k-ppl), we can use the data mixture as features and the avg(k-ppl) as target labels to fit a regression model.

### 3.4 FIT A REGRESSION MODEL USING THE DATA MIXTURES AS FEATURES AND THE AVG(K-PPL) AS THE TARGET LABEL

The next step is to fit a regression model using the data mixture ratios of three domains as features and the avg(k-ppl) as target labels. We choose the regression model (Liu et al., 2024) over the scaling law (Ye et al., 2024a) to forecast appropriate data mixture ratios for large-scale LLMs due to a lack of straightforward evaluation metrics for our problem. With the regression model, we are able to leverage a limited set of small models to estimate continuous target values for unknown data mixture ratios.

Specifically, considering the input features correspond to the domain ratios of the data mixture $R^n = [r_{general}^n : r_{code}^n : r_{math}^n]$, and the target label $\overline{k - ppl}_{PT}^n(Y_{RL}) \in PPL_{M_S}$, the goal is to find a function that best maps the input features to the target variable.

Taking linear regression as an example, which assumes a linear relationship between the input features and the target variable, can be represented as:

$$\overline{k - ppl}_{PT}^n(Y_{RL}) = \omega_0 + \omega_1 r_{general}^n + \omega_2 r_{code}^n + \omega_3 r_{math}^n \tag{6}$$

Where $\omega_0$ represents the error or noise in the data, and $\omega = (\omega_1, \omega_2, \omega_3)$ are the coefficients associated with the respective input features ($r_{general}^n : r_{code}^n : r_{math}^n$). The coefficients $\omega$ are typically estimated using techniques such as ordinary least squares, aiming to minimize the sum of squared residuals between the predicted and actual target values.

Alternative regression algorithms such as LightGBM (Ke et al., 2017) or SVR (Brereton & Lloyd, 2010) can also be used for fitting the regression models. Using known training data mixtures, these regression models can learn a predictive function that can estimate target values for arbitrary data mixtures without requiring additional training.

Once the regression model is fitted, we can efficiently explore the entire space of possible data mixtures. By leveraging the regression model to predict target values for each potential mixture, we rapidly identify the optimal input features $R^{best} = [r_{general}^{best} : r_{code}^{best} : r_{math}^{best}]$ that yield the lowest $\overline{k - ppl}_{PT}^n(Y_{RL})$ within an infinite set $PPL_{M_S}$:

$$[r_{general}^{best} : r_{code}^{best} : r_{math}^{best}] = \underset{R^n \in Set(R)}{argmin} (\overline{k - ppl}_{PT}^n(Y_{RL})) \tag{7}$$

This simulation-based optimization significantly reduces costs, as both small-scale model simulations and regression predictions are computationally inexpensive.

### 3.5 TRAIN LARGE-SCALE MODELS USING THE PREDICTED OPTIMAL DATA MIXTURE

After simulating and predicting the optimal data mixtures via the regression model, we apply the predicted top-P mixture ratios to sample large-scale pre-training data:

$$Set(R^{topP}) = [R_{top}^1, ..., R_{top}^P] \tag{8}$$

According to the hypothesis proposed and proved by Liu et al. (2024), the rank invariance of data mixtures posits that the relative ranking of data mixtures in terms of their impact on model performance remains consistent across different model scales and training token quantities. Thus, we train a larger model using these optimally mixed data, going through both pre-training and RL phases:

$$Set(M_L^{topP}(\theta_{PT})) = [M_L^1(\theta_{PT}^1), ..., M_L^P(\theta_{PT}^P)]$$
$$Set(M_L^{topP}(\theta_{RL})) = [M_L^1(\theta_{RL}^1), ..., M_L^P(\theta_{RL}^P)] \tag{9}$$

We can subsequently employ these well-trained larger models to perform the corresponding downstream tasks.

## 4 EXPERIMENTS

### 4.1 EXPERIMENTAL SETUPS

**Datasets and benchmarks.** For the pre-training stage, we train on a mixture of three datasets for code, math, and general corpus separately: TheStack (Kocetkov et al., 2022), OpenWebMath (Paster et al., 2023), and the subset C4 dataset of redpajama (Together, 2023). During the reinforcement learning fine-tuning phase, we train on instruction datasets such as python-code-25k (FlyTech, 2024), OMNI-MATH (Gao et al., 2024) and GSM8K (Cobbe et al., 2021) to further optimize the model. Then, we choose the following representative downstream benchmarks used by LLMs to assess the model's broad generalization capacity. General tasks: OpenBookQA(OBQA)(Mihaylov et al., 2018), PIQA(Bisk et al., 2020) and Hellaswag(Zellers et al., 2019). Coding tasks: HumanEval (Chen et al., 2021) and MBPP (Austin et al., 2021). Math tasks: MATH (Hendrycks et al., 2021) and GSM8K (Cobbe et al., 2021). The adoption of these datasets and benchmarks is aligned with the publicly released LLMs' technical reports (Dubey et al., 2024; M-A-P, 2024; Guo et al., 2025).

**Metrics.** For the regression model, we follow Liu et al. (2024) and employ two metrics to evaluate the regression model: the Spearman rank correlation coefficient ($\rho$) and mean squared error (MSE). The Spearman correlation $\rho$ provides a nonparametric measure of the strength and direction of association between two variables, while MSE evaluates the regression model's performance for small-scale proxy models by measuring the average squared difference between predicted and actual values. For the RL-trained LLMs, each downstream task is evaluated using its corresponding standard evaluation metrics.

**Baselines.** We compare Go4RL with four baselines: *Human*, *PPL*, *RL_loss*, and *Worst*. *Human* employs manual ratio allocation for data across the three domains based on empirical training experience according to the release of LLMs' technical reports (Dubey et al., 2024; M-A-P, 2024). Specifically, the mixing ratio for code, math, and general is 0.33:0.15:0.52. *PPL* uses the perplexity filtering methods from Ankner et al. (2024), selecting pretraining data by evaluating the model's perplexity on candidate data samples. *RL_loss* replaces the regression model's fitting target of avg(k-ppl) with the loss of the RL phase. *Worst* uses top-5 worst data mixtures selected by Go4RL.

**Implementation details.** For small-scale proxy models, we train a total number of 512 Transformer-based decoder-only language models with 1M parameters for 1B tokens, calculating their avg(k-ppl) scores to fit a regression model. We then validate the regression model on 256 unseen data mixtures across pretrained language models with 1M, 60M, and 1B parameters. Finally, we train and evaluate the 1B-parameter model on 64 unseen data mixtures with a total of 25B tokens. Following the methodology of Liu et al. (2024), we scale the token distribution by values ranging from 0.1 to 5.0 to construct a wide spectrum of sparse and near-uniform distributions. These scaled distribution vectors are then used as the hyperparameters $\alpha$ for the Dirichlet sampling. We consider LightGBM, ridge regression, and random forest as our regression models.

We pre-train the decoder-only base models with 1 epoch, using the Adam optimizer in a batch size of 128K with a learning rate of 4e-4. The weight decay is 0.005. For the reinforcement learning stage, we employ both representative online (GRPO (Shao et al., 2024) following Deepseek-r1 (Guo et al., 2025)) and offline (DPO (Rafailov et al., 2023)) RL algorithms. Due to space constraints, the main text presents part of the experimental results, while the other experimental results are provided in the Appendix. We guide the learning of the model by quantifying the quality of its answers, scoring based on the output format and the final answer. $k$ responses for calculating the avg(k-ppl) is set to 8. Our setup consists of a four-core CPU and eight NVIDIA Tesla A100 GPUs.

### 4.2 THE PERFORMANCE OF THE REGRESSION MODEL

We first evaluate the ability of the regression model, including the fitting results over different regression algorithms, and the performance in predicting the effect of unseen data mixtures.

**(1) Fitting results over different regression algorithms.** We fit the regression model using small proxy models with 1M parameters and evaluate the loss prediction performance on small models with the same parameters. As illustrated in Table 1, we adopt three representative regression algorithms: Light GBM, Ridge Regression, and Random Forest. LightGBM surpasses the other two

regression methods in predicting the unseen data mixing ratio over the same 1M parameter models, achieving 86.46% in the $\rho$ metric and 0.34 in the MSE metric.

Table 1: We separately fit three regression models based on the results of the 1M parameter models trained on 1B tokens and validate them on unseen data mixtures for 1M, 60M, and 1B parameter models. The 60M parameter model is trained with 1B tokens, while the 1B model is trained with 25B tokens. $\rho$ compares the predicted and actual data mixtures, while MSE measures the differences between the target performances of the predicted and actual models.

| Model Size | 1M | | 60M | 1B |
|---|---|---|---|---|
| Metrics | $\rho$ | MSE | $\rho$ | $\rho$ |
| LightGBM | 86.46 | 0.34 | 69.12 | 58.33 |
| Ridge Regression | 83.12 | 0.37 | 61.65 | 51.67 |
| Random Forest | 73.24 | 0.45 | 38.8 | 8.33 |

The regression models are then used to predict models with scaled-up parameters and tokens. The prediction for larger models with more tokens demonstrates that regression models trained on small proxy models can be applied to larger-scale models. We train different 60M and 1B parameter LLM models (both pre-trained and RL-trained) to evaluate the deviation between the predicted and actual results for unseen data mixtures.

Table 1 shows that the regression models of LightGBM and Ridge Regression performed reasonably well across all three model scales. The random forest method fails to predict scaled-up models, which may be because the tree-structure-based algorithm could not learn the complex internal pattern of the pre-training and RL-training pipeline. While prediction accuracy declined as model size increased for all three regression models, the Spearman rank correlation consistently remain around 70% and 60% for LightGBM. The variance in prediction performance of parameter-scaled models may be attributed to shifts in data samples from a single data source, particularly the training tokens scaled for larger models, e.g., 25b tokens for 1B parameter models. Among these, LightGBM exhibits the most outstanding performance, outperforming the RidgeRegression by around 8% points. These results validate the feasibility of the rank-invariance-based approach for predicting optimal data mixtures.

**(2) Deep analysis of the fitted LightGBM performance on predicting the data mixture.** To further analyze the distribution of the top-P mixtures, we separately visualize their weight distributions across all three domains (general, code, and math). As shown in Figure 2(b) with P=50, we observe that top-ranked data mixtures consistently contained a higher proportion of domain code up to the range of 70% to 80%, while the proportion of general text stays close to 20%. This phenomenon aligns with the findings reported in Zhao et al. (2025)'s study.

### 4.3 THE IMPACT OF PREDICTED MIXTURE ON DOWNSTREAM TASKS

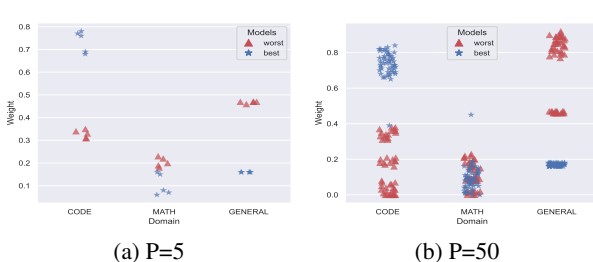

(a) P=5          (b) P=50

Figure 2: The distribution of Top-P (P=5, 50) best and worst mixtures across three domains.

We validate the efficacy of Go4RL on downstream reasoning tasks. The selected benchmarks are prevalently adopted by released LLMs (Dubey et al., 2024; Guo et al., 2025) and comprehensively cover tasks across three domains: code, mathematics, and general reasoning, enabling a holistic evaluation of Go4RL's capabilities. We compare the Go4RL predicted data mixture with four baselines: HUMAN, PPL, Worst, and RL_loss. Refer to the details of these baselines in experimental setups. We first utilize Go4RL to predict the top 5 best and worst data mixture strategies. We train 1B parameter base models using the predicted data mixture and then use both GRPO and DPO to get RL-trained models separately.

Table 2 lists the average performance of 1B models trained by GRPO, using the top-5 best data mixtures selected by Go4RL. The results with DPO are listed in the Table 5 in appendix due to the page limit. According to Table 2, pretraining data mixtures using the proposed optimization objective avg(k-ppl) have an obvious impact on reasoning downstream tasks. Compared to the four baselines, Go4RL demonstrates a superior performance, outperforming all four competing baselines in 4 out of 7 tasks while achieving the highest average score (3.8% higher than RL_loss). The average performance improvement across all the tasks highlights both the crucial role of data mixtures in the pretraining+RL pipeline and the validity of our data selection method. Note that the RL_loss baseline shows suboptimal performance, indicating that the RL loss function alone is not an adequate metric for the pre-train+RL pipeline. We also notice the performance decline on GSM8K. The potential reason may be its data format deviating significantly from the training data OpenWebMath. Similar phenomena have been found in other works (Zhao et al., 2025). Besides, the settings of our RL stage comprise GSM8K training data. The performance decay may reflect prior research of the RL narrowing the reasoning bounds of the base model (Yue et al., 2025). This also indicates the complexity of the relationship between pre-training and RL, which needs further exploration.

Table 2: Average performance of 1B models trained by GRPO, using the top-5 best data mixtures selected by Go4RL, compared with several other data selection baselines. BSET ON indicates how many of all tasks the model achieves the best results in. The reported performance on each task is the average score of 3 runs, while the highest score on each task is highlighted in bold.

| | Domain | General | | | Code | | Math | | | |
|---|---|---|---|---|---|---|---|---|---|---|
| | Tasks | OBQA | PIQA | Hellaswag | Humaneval | MBPP | MATH | GSM8K | Avg. | Best On |
| | Metrics | Acc. | Acc. | Acc. | Pass@10 | Pass@10 | Acc | Acc | | |
| Baselines | HUMAN | 17.8 | 14.3 | 22.9 | 7.3 | 0.6 | 0.4 | 0.5 | 9.1 | 0 |
| | PPL | **18.8** | 17.2 | 24.6 | 9.2 | 1.4 | 0.6 | **2.8** | 10.7 | 2 |
| | RL_loss | 5.8 | **24.5** | 23.0 | 20.1 | 6.0 | 0.9 | 1.9 | 11.7 | 1 |
| Go4RL | Worst | 18.5 | 16.7 | 25.1 | 16.9 | 2.7 | 0.6 | 0.3 | 11.5 | 0 |
| | Best | 12.5 | 21.2 | **25.3** | **36.8** | **10.5** | **1.1** | 0.9 | **15.5** | 4 |

We further plot the top-5 best data mixture ratio distributions as illustrated in Figure 2(a). The best data mixture strategies have a relatively high ratio on the code domain and a similar low ratio on both the general and math domains. However, the downstream tasks show performance gain over all the domains.

### 4.4 HOW DOES THE PERFORMANCE OF MODELS WITH DIFFERENT DATA MIXTURES IMPROVE DURING RL?

We are interested in how the RL-trained models favor the domain distribution with the accumulated RL steps. As shown in Figure 3, we compare trajectories during RL of the models pretrained with data mixtures selected by Go4RL (top-5 best) with the random sampling, Go4RL-worst, and RL_loss under the DPO algorithm. For the code domain, we compare the pass@k (k=1,10) trajectories over MPBB and Humaneval. Go4RL-best demonstrates strong RL gains from the starting point on Humaneval, eventually achieving the largest RL reasoning gains for both the pass@1 and pass@10 evaluations on all the Code tasks. For both MATH and GSM8K, Go4RL-best demonstrates a clear advantage during RL. It can be observed that the uptrend of the RL gains for the MPBB and MATH benchmarks with the Go4RL-optimized data mixture. For many benchmarks, we noticed that the RL_loss baseline displays initial performance gains while fluctuating as the RL steps accumulate. These results further reflect the recent findings of the entropy collapse phenomenon for RL phase (Yu et al., 2025). It also proves that the performance in the RL phase is not the sole factor affecting downstream knowledge expression and generation, finding a suitable base model also plays a significant role. The results with GRPO are listed in Figure 7 in the Appendix.

### 4.5 WHAT'S A PROPER MEASUREMENT FOR THE DATA MIXTURES TARGETING RL PERFORMANCE?

We want to further check if the proposed avg(k-ppl) is proper for the performance measurements by evaluating its correlation with the data mixture ratios. It should be noted that there are other options, such as the downstream task accuracy and RL loss. RL loss has been proven to be inadequate in the

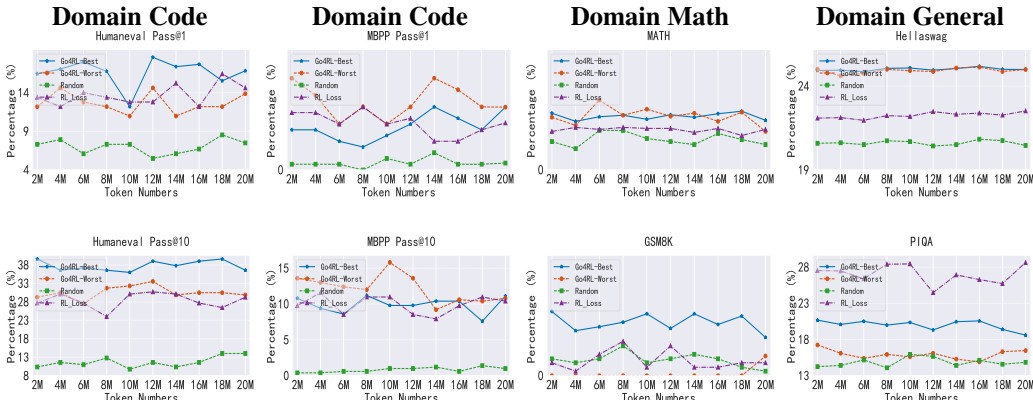

Figure 3: Three domains' downstream performances of the Go4RL-optimal 1B parameter models (RL-trained with DPO) with top-5 best data mixtures, compared with the random sampling, RL_loss and Go4RL selected worst baselines. The x-axis is the growing training tokens in the RL stage.

previous analysis. We visualize the correlation between the avg(k-ppl), the potential measurement of downstream task accuracy, and the different data domains under the GRPO.

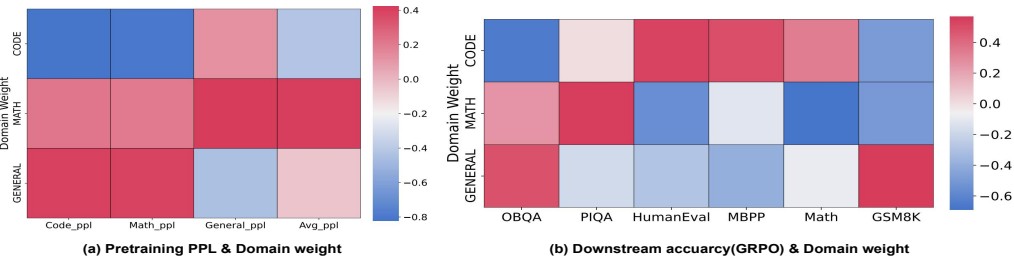

Figure 4: (a) The correlations between avg(k-ppl) and training data ratios across three domains separately and averaged over 1B models. (b) The correlations between downstream accuracy and training domain ratios on 1B models.

According to Figure 4(a), illustrating the correlation during pre-training stages, the avg(k-ppl) elaborates a high correlation with the data domain ratios, demonstrating its effectiveness in evaluating a suitable base model for RL. In the pre-training stage, data from the math domain provides gains to all other domains. The general domain significantly influences both code_ppl and math_ppl, which are calculated as the avg(k-ppl) from the code and math domains separately. For the downstream tasks illustrated in Figure 4(b), the data from the code source exhibits a pronounced negative impact on OBQA, which gives an explanation of the OBQA performance decline and performance of code benchmarks increase in Table 2. The visualization results highlight the importance and inherent complexity of determining the optimal pre-training data mixture and give hints for the further improvement of the Go4RL method.

## 5 CONCLUSION

On the basis of the recent research findings, we propose a framework Go4RL, to predict optimal data mixture strategies that effectively make the base model suitable for augmenting RL's reasoning capabilities. Leveraging the ratios of data from different domains as features and the proposed average perplexity of the responses generated by the RL-trained models as the target, Go4RL fits a regression model to predict the optimal data mixture leading to enhanced RL capabilities. We conduct experiments to validate Go4RL's performance. Our work gives a first insight into how the pre-trained data mixture of the base model further impacts the RL models that could potentially inspire broad discussion in the research community.

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

## A  THE USE OF LARGE LANGUAGE MODELS (LLMs)

We appreciate the assistance provided by GPT-4 (Achiam et al., 2023) in writing aid and sentence-level polishing.

## B  RL ALGORITHM DETAILS

### B.1  PRELIMINARY

We model a LLM as a policy $\pi_\theta(y|x)$ parameterized by $\theta$, where $\pi_\theta$ generates a text response $y \in Y$ conditioned on a user instruction $x \in X$. Given a prompt $x$, the LLM $\pi_\theta$ will generate response $y$ in an auto-regressive manner:

$$\pi_\theta(y|x) = \prod_t \pi_\theta(y_t|x, y_{<t}) \tag{10}$$

where $y_t$ denotes the $t$-th token in the response, and $y_{<t}$ represents the sequence of tokens preceding $y_t$.

**RLHF**: To further align the SFT model $\pi_\theta$ with human preferences, prior studies introduce the Reinforcement Learning from Human Feedback (RLHF) framework (Ouyang et al., 2022), which optimizes the following objective:

$$\mathcal{L}_r(\pi_\theta) = \mathbb{E}_{x \sim p_{data}, y \sim \pi_\theta}[r(x, y) - \beta log \frac{\pi_\theta(y|x)}{\pi_{ref}(y|x)}] \tag{11}$$

where $r$ denotes the reward function that captures human preferences, mapping a prompt–response pair to a scalar value. $\pi_{ref}$ is the reference model used to regularize $\pi_\theta$ via Kullback–Leibler divergence, and $\beta$ is a constant that controls the strength of this regularization.

## B.2 DPO

A line of RL works aims to skip the reward modeling step and may be referred to as the direct preference learning approach (Tang et al., 2024). Among them, the direct preference optimization (DPO) algorithm is the most popular one. DPO (Rafailov et al., 2023) is a preference-based alignment algorithm that directly optimizes the language model toward human-preferred outputs without relying on reinforcement learning. Instead of training a reward model as in RLHF, DPO reformulates preference learning as a supervised objective over pairs of responses.

Specifically, it leverages pairs of preferred and dispreferred responses and maximizes the log-likelihood of the preferred output relative to the dispreferred one under the model's distribution. Given a prompt $x$ with two candidate responses $(y^+, y^-)$, where $y^+$ is preferred over $y^-$ according to human feedback, DPO optimizes the policy $\pi_\theta$ by maximizing the following objective:

$$\mathcal{L}_{DPO}(\theta) = \mathbb{E}_{(x, y^+, y^-)}[log\sigma(\beta(log \frac{\pi_\theta(y^+|x)}{\pi_{ref}(y^+|x)} - log \frac{\pi_\theta(y^-|x)}{\pi_{ref}(y^-|x)}))] \tag{12}$$

where $\pi_\theta$ is the policy being optimized, $\pi_{ref}$ is a frozen reference model, $\beta$ is a temperature parameter controlling the sharpness of preference weighting, and $\sigma(\cdot)$ is the logistic sigmoid function.

DPO allows direct optimization of the model distribution with respect to preferences, avoiding the instability of reinforcement learning while maintaining strong alignment performance.

## B.3 GRPO

Group Relative Policy Optimization (GRPO) is a reinforcement learning variant proposed in Shao et al. (2024). It is designed for fine-tuning large language models (LLMs) in reasoning-heavy tasks, and aims to reduce computational cost and complexity while maintaining high performance. Unlike Proximal Policy Optimization (PPO), GRPO removes the need for a separate value function (critic), instead it estimates the baseline using group-based statistics.

Given a prompt $x$, the old policy $\pi_{\theta_{old}}$ samples a group of $G$ candidate responses $\{y_1, y_2, ..., y_G\}$. Each response $y_i$ is evaluated by a reward function $r(x, y_i)$, yielding scores $\{r_1, r_2, ..., r_G\}$. The group statistics are then computed as:

$$\mu = \frac{1}{G}\sum_{i=i}^{G} r_i, \sigma = \sqrt{\frac{1}{G}\sum_{i=i}^{G}(r_i - \mu)^2} \tag{13}$$

A normalized group-relative advantage is defined for each response as:

$$A_i = \frac{r_i - \mu}{\sigma} \tag{14}$$

Let the importance sampling ratio be:

$$\rho_i = \frac{\pi_\theta(y_i|x)}{\pi_{\theta_{old}}(y_i|x)} \tag{15}$$

The GRPO objective extends the PPO surrogate loss with group-relative advantages and KL regularization to a reference model $\pi_{ref}$:

$$\mathcal{L}_{GRPO}(\theta) = \mathbb{E}_{(x,y_1,\ldots,G)\sim\pi_{\theta_{old}}}[\frac{1}{G}\sum_{i=1}^{G}min(\rho_i A_i, clip(\rho_i, 1-\varepsilon, 1+\varepsilon)A_i) - \beta D_{KL}(\pi_\theta(\cdot|x)||\pi_{ref}(\cdot|x))]$$

(16)

where $\varepsilon$ is the clipping parameter that constrains policy updates, and $\beta$ controls the strength of KL regularization.

GRPO eliminates the need for an explicit value function, replacing it with group-relative statistics as the baseline, thereby simplifying optimization while maintaining stability.

## C DETAILS OF MODELS AND SAMPLES

### C.1 MODEL DETAILS

The structures of the series of models are displayed in Table 3. These models utilize the same vocabulary as Qwen Yang et al. (2025), with a vocabulary size of 151,936.

| Model | 1M | 60M | 1B |
|---|---|---|---|
| **Vocabulary Size** | 151936 | 151936 | 151936 |
| $n_{layers}$ | 2 | 10 | 22 |
| $n_{heads}$ | 8 | 8 | 16 |
| $n_{embedding}$ | 256 | 768 | 2048 |
| $d_{model}$ | 512 | 1536 | 5632 |

Table 3: The model structure from different LLMs.

### C.2 SAMPLE ANALYSIS

In the main paper, we hypothesize that the performance drop on GSM8K may stem from its significant deviation in data format compared to the training data, OpenWebMath. To further substantiate this claim, we extract and compare sample instances from both datasets to illustrate their differences. OpenWebMath and GSM8K differ significantly in format, content, and purpose. OpenWebMath is a large-scale, web-crawled mathematical text corpus designed for pretraining, containing a mix of continuous mathematical content (e.g., LaTeX equations, proofs). In contrast, GSM8K is a curated dataset of grade-school-level math word problems, formatted as question-answer pairs with explicit step-by-step reasoning.

Table 4 shows the significant differences between OpenWebMath and GSM8K. The performance degradation on GSM8K may be attributed to the following potential factors based on our analysis:

- Pretraining on OpenWebMath may inadequately prepare models for GSM8K's task-specific format.
- OpenWebMath's focus on declarative mathematical knowledge might not transfer to GSM8K's procedural reasoning demands.
- Models pretrained on continuous text may struggle with GSM8K's segmented Q&A structure, lacking fine-tuned alignment for answer generation.

We use a heatmap to show the relationship between the ratios of pre-training data sources (general, code, and math) and downstream tasks under GRPO. As illustrated in Figure 5, it can be clearly observed that increasing the ratios of Math to General, Code to General, and Code to Math data leads to positive effects on most tasks except for the Math-related tasks. In other words, a higher proportion of math-domain pre-training data tends to degrade performance on the GSM8K task. This observation supports our hypothesis that there exists a noticeable gap between OpenWebMath (which is more oriented toward theorems and proofs) and GSM8K&Math (which focuses more on arithmetic operations such as addition and subtraction) in terms of data format and knowledge coverage.

| Dataset | | Sample |
|---|---|---|
| **OpenWebMath** | Question: | What is the derivative of $f(x) = (e^{2x})(ln(x))$? |
| | Answer: | $f'(x) = e^{2x}\left(2\ln x + \frac{1}{x}\right)$ 
 Explanation: The derivative of $\ln x$ is $\frac{1}{x}$ 
 The derivative of $e^g(x)$ is $e^g(x) \cdot g'(x)$ 
 The derivative of $h(x) \cdot l(x)$ is $h'(x) \cdot l(x) + h(x) \cdot l'(x)$ 
 Then $f'(x) = e^{2x} \cdot 2 \cdot \ln x + e^{2x} \cdot \frac{1}{x} = e^{2x}\left(2\ln x + \frac{1}{x}\right)$ |
| **GSM8K** | Question: | Bella has two times as many marbles as frisbees. She also has 20 more frisbees than deck cards. 
 If she buys $2/5$ times more of each item, what would be the total number of the items she 
 will have if she currently has 60 marbles? Let's think step by step. |
| | Answer: | When Bella buys $2/5$ times more marbles, she'll have increased the number of marbles by $2/5 * 60 = 24$. 
 The total number of marbles she'll have is $60 + 24 = 84$. If Bella currently has 60 marbles, and she has 
 two times as many marbles as frisbees, she has $60/2 = 30$ frisbees. If Bella buys $2/5$ times more frisbees, 
 she'll have $2/5 * 30 = 12$ more frisbees. The total number of frisbees she'll have will increase to $30 + 12 = 42$. 
 Bella also has 20 more frisbees than deck cards, meaning she has $30 - 20 = 10$ deck cards. If she buys $2/5$ times 
 more deck cards, she'll have $2/5 * 10 = 4$ more deck cards. The total number of deck cards she'll have is $10 + 4 = 14$. 
 Together, Bella will have a total of $14 + 42 + 84 = 140$ items. The answer is 140. |

Table 4: Samples from OpenWebMath and GSM8K.

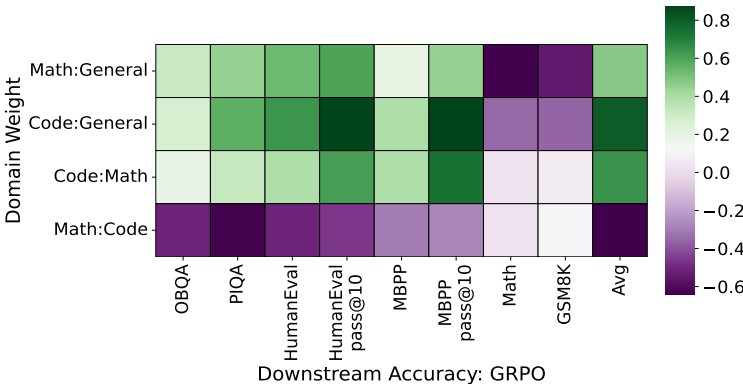

Figure 5: The relationship between the ratios of pre-training data sources (general, code, and math) and downstream tasks under GRPO.

# D  SUPPLEMENTARY EXPERIMENTS

## D.1  ANALYSIS OF THE FITTED LIGHTGBM PERFORMANCE ON PREDICTING THE DATA MIXTURE

We further explore the performance of the best-performing regression model (LightGBM) across a broader space, where the training data is generated based on GRPO RL algorithm. Figure 6 illustrates the simulated new data mixtures alongside the corresponding predicted targets to find the best data mixture. As can be seen from the figure, when the regression model achieves its optimal target value, the proportions of code and math data are relatively low.

## D.2  THE IMPACT OF PREDICTED MIXTURE ON DOWNSTREAM TASKS UNDER THE DPO ALGORITHM

We further validate the efficacy of Go4RL on downstream reasoning tasks under DPO. We compare the Go4RL predicted data mixture with three baselines: Random, RL_loss and Worst. Random randomly selects five distinct data mixtures to train models following the pre-training + RL pipeline, and the average performance of these five models is reported as the final result. Refer to the details of other baselines in experimental setups. At first, we utilize Go4RL to predict the top 5 best data mixture strategies. Then We train 1B parameter-based models using the predicted data mixture and use DPO to get RL-trained models separately. Table 5 lists the average score of 3 runs for the top-5 best 1B RL-trained models selected by Go4RL.

Table 5 shows that pretraining data mixtures using the proposed optimization objective avg(k-ppl) have an obvious impact on reasoning downstream tasks. Compared to the three baselines, Go4RL

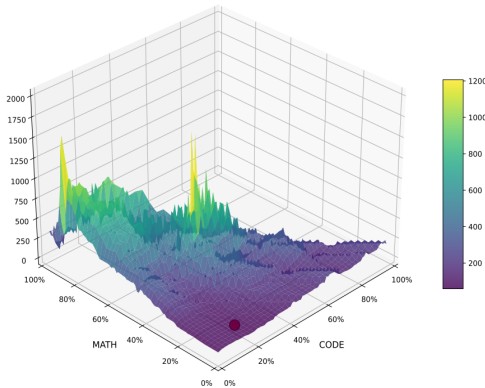

Figure 6: Simulate potential data mixtures across a broader space and use the regression model to predict the best data mixture (lowest avg(k-ppl) marked with the red dot).

Table 5: Average performance of 1B models trained by DPO, using the top-5 best data mixtures selected by Go4RL, compared with several other data selection baselines. BSET ON indicates how many of all tasks the model achieves the best results in. The reported performance on each task is the average score of 3 runs, while the highest score on each task is highlighted in bold.

|  | Domain | General | | | Code | | Math | | Avg. | Best On |
|---|---|---|---|---|---|---|---|---|---|---|
|  | Tasks | OBQA | PIQA | Hellaswag | Humaneval | MBPP | MATH | GSM8K | | |
|  | Metrics | Acc. | Acc. | Acc. | Pass@10 | Pass@10 | Acc | Acc | | |
| Baselines | Random | 7 | 14.8 | 20.5 | 14.0 | 1.0 | 0.5 | 0.1 | 8.3 | 0 |
|  | Human | 8.6 | 6.9 | 22.5 | 20.1 | 2.8 | 0.6 | **2.0** | 9.1 | 1 |
|  | RL_loss | 7.6 | **28.6** | 22.5 | 29.3 | 10.4 | 0.8 | 0.2 | 14.2 | 1 |
| Go4RL | Worst | **18.6** | 16.5 | **24.5** | 29.9 | 10.8 | 0.7 | 0.4 | 14.5 | 2 |
|  | Best | 12.5 | 18.6 | **24.5** | **36.6** | **11.1** | **1.0** | 0.7 | **15.0** | 4 |

demonstrates a superior performance, outperforming all three competing baselines in 4 out of 7 tasks while achieving the highest average score. The average performance improvement across all the tasks highlights both the crucial role of data mixtures in the pretraining+RL pipeline and the validity of our data selection method. The RL_loss baseline also exhibits suboptimal performance under DPO, indicating that the RL loss function alone is insufficient as an evaluation metric for the pre-train+RL pipeline. These results from both GRPO and DPO demonstrate that Go4RL serves as a versatile framework for selecting data mixture ratios. Regardless of the reinforcement learning algorithm used, models trained with Go4RL consistently achieve superior average performance across downstream tasks in various domains.

We also observe a performance decline on general tasks. A potential explanation is that the training data used in the RL phase exhibits a stronger bias toward in-domain content, thereby narrowing the reasoning capacity of the base model, which is a phenomenon consistent with prior research (Yue et al., 2025). This also indicates the complexity of the relationship between pre-training and RL, which needs further exploration.

### D.3 HOW DOES THE PERFORMANCE OF MODELS WITH DIFFERENT DATA MIXTURES IMPROVE DURING RL UNDER GRPO?

We are interested in how the RL-trained models favor the domain distribution with the accumulated RL steps. As shown in Figure 7, we compare trajectories during RL of the models pretrained with data mixtures selected by Go4RL (top-5 best) with the random sampling, RL_loss and Go4RL-worst baselines. For the code domain, we compare the pass@k (k=1,10) trajectories over MPBB and Humaneval benchmarks. Go4RL-best demonstrates strong RL gains from the starting point, eventually achieving the largest RL reasoning gains for almost all the pass@1 and pass@10 evaluations. For the MATH benchmark, Go4RL-best consistently maintains a significant lead throughout the training process. On the PIQA benchmark, Go4RL and RL_Loss achieve comparable performance, with both methods consistently maintaining a leading position.

Like the results of DPO, these results also prove that the performance in the RL phase is not the sole factor affecting downstream knowledge expression and generation. Finding a suitable base model also plays a significant role.

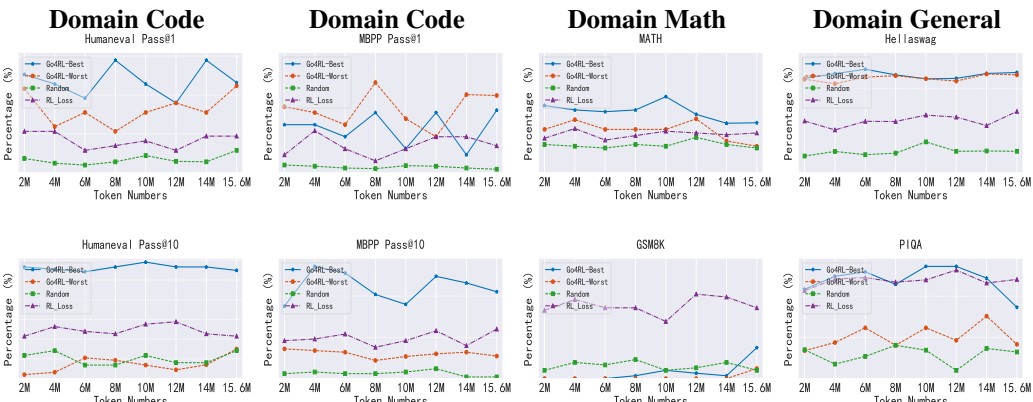

Figure 7: Three domains' downstream performances of the Go4RL-optimal 1B parameter models with top-5 best data mixtures, compared with the random sampling method, RL_loss and Go4RL-worst baselines under GRPO algorithm. The x-axis is the growing training tokens in the RL stage.

## D.4 WHAT'S A PROPER MEASUREMENT FOR THE DATA MIXTURES TARGETING RL PERFORMANCE?

In addition, we visualize the correlation between the potential measurement of downstream task accuracy and the different data domains under the DPO.

For the downstream tasks illustrated in Figure 8(b), beyond general-domain benchmarks such as OBQA and PIQA, the proportion of code data exhibits a positive correlation with performance on most downstream tasks. This outcome is particularly evident on the HumanEval benchmark, which aligns well with our intuitive expectations. An increase in the proportion of math data proves beneficial to the performance on the OBQA task to some extent, demonstrating that enhanced mathematical capability can effectively improve the model's logical reasoning and structured expression abilities. The visualization results highlight the importance and inherent complexity of determining the optimal pre-training data mixture and give hints for the further improvement of the Go4RL method.

The visualization results highlight the importance and inherent complexity of determining the optimal pre-training data mixture and give hints for the future improvement of the Go4RL method.

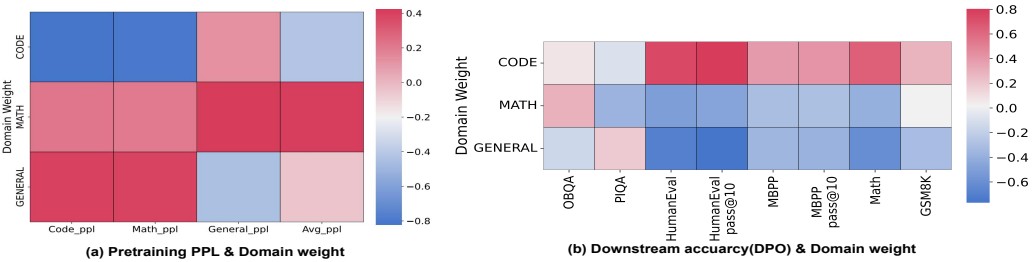

Figure 8: (a) The correlations between avg(k-ppl) and training data ratios across three domains separately and averaged over 1B models. (b) The correlations between downstream accuracy and training domain ratios on 1B models.

### D.5 IS IT WORTH USING THE PROXY MODELS?

We report the GPU hours and FLOPs required to fit 512 1M-parameter models for predicting the optimal data mixtures, and compare them against other baselines. As listed in the Table 6, the computational cost of our approach remains within an acceptable and marginal range.

Our target is to predict the optimal data mixture for larger base models, thereby saving the cost of retraining the base models to get the optimal one. When compared to the entire cost of training a 1B base model on 25B tokens, training proxy models is rather inexpensive—only roughly 5%. The effect would be significantly greater when the training data and base model parameters are increased. Therefore, the choice of proxy models should be viewed as a trade-off between computational efficiency and final model performance.

Table 6: Computational consumption of Go4RL, compared with other baselines.

| Method | PPL | RL/_loss | Human | Go4RL | | |
|---|---|---|---|---|---|---|
| Stage | Pre-training | Total | Pre-training | Pre-training | RL | Total |
| FLOPs | 1.95E+19 | 7.45E+19 | 0 | 3.07E+18 | 7.14E+19 | 7.45E+19 |
| GPU hours | 4.08 | 16.14 | 0 | 0.64 | 15.5 | 16.14 |

### D.6 WHY IS AVG(K-PPL) SUPERIOR TO OTHER FITTING TARGETS?

**RL_loss**: We further analyze the correlation between avg(k-ppl) and RL_loss with downstream tasks, as well as the correlation between avg(k-ppl) and RL_loss themselves, to substantiate why we choose avg(k-ppl) as the fitting target.

The higher the correlation between metrics, the larger the absolute value. As shown in Figure 9, choosing avg(k-ppl) as the target for fitting yields noticeably greater advantages and gains on downstream tasks compared to directly fitting the RL loss ($|-0.57| > |-0.37|$). Additionally, the positive correlation between avg-ppl and RL loss ($value = 0.35$) indicates that avg(k-ppl) and RL_loss influence the model in the same direction. These empirical factors make avg(k-ppl) our first choice for fitting the regression model.

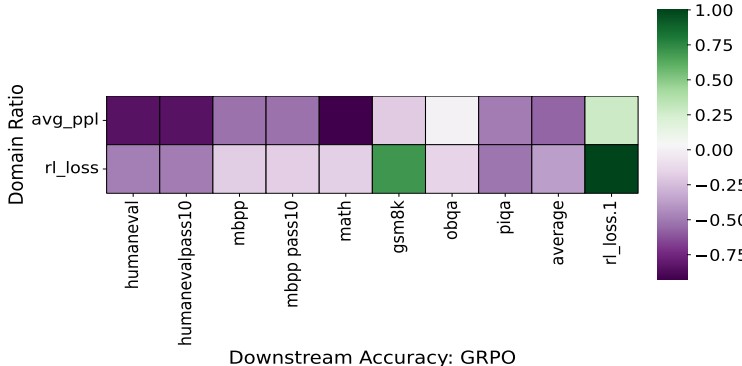

Figure 9: The correlations between avg(k-ppl) and RL_loss with downstream tasks over 1B models, under GRPO algorithm.

**avg(k-entropy)**: We also conduct experiments with the same settings to compare the fitting targets avg(k-entropy) with avg(k-ppl). The experimental results are listed in Table 7. According to the results, avg(k-ppl) surpasses the entropy metrics in downstream tasks.

**downstream metrics**: Using the downstream metrics, like the pass@k metric could also evaluate the performance. However, we argue that using downstream-task pass@k as the regression target may introduce several inherent limitations:

Table 7: Average performance of 1B models trained by GRPO, using the top-5 best data mixtures selected by Go4RL, compared with avg(k-entropy). BSET ON indicates how many of all tasks the model achieves the best results in. The reported performance on each task is the average score of 3 runs, while the highest score on each task is highlighted in bold.

| Domain | General | | | Code | | Math | | | |
|---|---|---|---|---|---|---|---|---|---|
| Tasks | OBQA | PIQA | Hellaswag | Humaneval | MBPP | MATH | GSM8K | Avg. | Best On |
| Metrics | Acc. | Acc. | Acc. | Pass@10 | Pass@10 | Acc | Acc | | |
| avg(k-entropy) | 16.2 | 10.6 | 24.9 | 4.3 | 0 | 0.2 | 2.3 | 8.4 | 2 |
| avg(k-ppl) (Go4RL) | 12.5 | **21.2** | **25.3** | **36.8** | **10.5** | **1.1** | 0.9 | **15.5** | 5 |

- High Noise and Instability: pass@k is inherently a discrete and high-variance metric, particularly on code, math, or multi-step reasoning tasks with limited evaluation samples, which injects significant noise into the regression targets and weakens the regressor's ability to learn stable performance trends.

- Strong nonlinearity with limited samples: The relationship between pass@k and the model's true capability is often highly nonlinear and even piecewise. The regression model may struggle to capture this structure, leading to underestimation of improvements in some regions or overestimation of gains in the high-performance regime.

- Lack of Generalizable Capability: Using pass@k as the regression target effectively ties the objective to a particular task, prompt format, and evaluation configuration. As a result, the inferred optimal data mixture may overfit to idiosyncrasies of that specific downstream task, and therefore may not generalize well to other tasks or to large-scale RL training.

### D.7 CAN GO4RL BE EXTENDED TO A LARGER MODEL?

To further validate the feasibility of applying Go4RL to larger-scale models, we train 300 proxy models with 60M parameters each on 1B tokens, fit a regression model, and use the regression model's predicted data mixtures to guide the training of a 3B parameter model.

As shown in Table 8, the regression model achieves 68.23% in the $\rho$ metric, demonstrating that Go4RL can scale effectively to larger models. When the target model becomes larger, using proxy models with more parameters leads to higher prediction accuracy in the fitted regression model, which aligns with our intuition. Therefore, selecting the proxy model size requires balancing computational budget against performance gains, depending on the practical constraints of the computational sources.

Table 8: We separately fit three regression models based on the results of the 60M parameter models trained on 1B tokens and validate them on unseen data mixtures for 60M and 1B parameter models. $\rho$ compares the predicted and actual data mixtures.

| | 60M>60M | 60M>3B |
|---|---|---|
| **Metric** | $\rho$ | $\rho$ |
| **lightGBM** | 77.85 | 68.23 |

### D.8 IS GO4RL STABLE UNDER DIFFERENT SAMPLING SETTINGS?

We conduct further experiments over different sampling settings, including k values and random seeds. As shown in Table 9, the variation of the settings shows that the $\rho$ values, which indicate regression performance, remain relatively stable.

Table 9: $\rho$ compares the predicted and actual data mixtures, under different sampling settings.

| Method | | $\rho(1M\text{->}1M)$ |
|---|---|---|
| **k-value** | k=32 | 87.82 |
| | k=128 | 86.46 |
| **random_seeds** | seeds = 1234 | 85.23 |
| | seeds = 4321 | 86.19 |

