# OpenReview forum: "Go4RL: Improving the Pre-training Data Mixture of Large Language Models for Enhancing Reinforcement Learning"
_ICLR.cc/2026/Conference — ICLR 2026 Conference Withdrawn Submission_

### Official Review · Reviewer_d5ci · 2025-10-17

**Soundness:** 2
**Presentation:** 2
**Contribution:** 2
**Rating:** 2
**Confidence:** 3

**Summary:**

the paper proposes a new way to select pretraining mixture that is beneficial for the model to perform downstream RL. The method uses the avg perplexity scores of the base model on RL generated k-responses as the metric to find the optimal mixing ratio.

**Strengths:**

- relevant topic, writing is mostly clear

**Weaknesses:**

- The method seems VERY expensive. it needs to train couply proxy models from pretrain to RL to even get a proxy to select data at the end. This could not be imagined to be deployed to large scale jobs such as training the next GPT.
- it is also unclear that proxy models can reliably predict the performance of a bigger sized model training on the same data. This is not a trivial transition
- evaluated benchmarks are very out of date

**Questions:**

See weakness

---

> ### Author Response · Authors · 2025-11-20
> **Rebuttal by Authors [1/1]**
>
> Thank you very much for your valuable comments and questions. We appreciate the time and effort you have put into reviewing our manuscript. Below, We address your concerns and provide further clarifications.
>
> ---
>
> > __W1__: The method seems VERY expensive. It needs to train couple proxy models from pretrain to RL to even get a proxy to select data at the end. This could not be imagined to be deployed to large-scale jobs such as training the next GPT.
>
> __A1__: We completely understand the review's concern with the computation cost of our method.
>
> Pls let us first explain that our target is to predict the optimal data mixture for larger base models, thereby saving the cost of retraining the base models to identify the optimal one. For example, __training proxy models is rather inexpensive—only roughly 5% of the total cost of training a 1B base model on 25B tokens__, according to our statistics listed below. We report the GPU hours and FLOPs required to fit 512 1M-parameter models for predicting the optimal data mixtures, and compare them against other baselines. The computational cost of our approach remains within an acceptable and marginal range. Therefore, the choice of proxy models should be viewed as a trade-off between computational efficiency and final model performance.
>
> |Methods	 | PPL | RL_loss |  HUMAN | | Go4RL(ours) | |
> | ---: |  :----:   |  :---: |   :----:   | :---: |   :----:   |  :---: |
> |__Stages__| Pre-training | Total | | Pre-training | RL | Total |
> | __FLOPs__ | 1.95E+19 | 7.45E+19 | 0 | 3.07E+18 | 7.14E+19 | 7.45E+19 |
> |__GPU hours__| 4.08 | 16.14 | 0 | 0.64 |15.5 | 16.14|
>
> Regarding the practical usability of the methods for large-scale jobs, we observed that the OLMO team's recent keynote at EMNLP 2025 demonstrated their use of regression methods to predict the optimal solution for pre-training OLMO series models (with different objective than our work). We hope this could prove that the regression-based method is promising in practice.
>
> We've also listed these results in the $\textrm{\color{blue}Appendix D.5}$ of the revised PDF.
>
> ---
>
> > __W2__: It is also unclear that proxy models can reliably predict the performance of a bigger sized model training on the same data. This is not a trivial transition
>
> __A2__: Thank you for your valuable question.
>
> To further validate the feasibility of applying Go4RL to larger-scale models, we trained 300 proxy models with 60M parameters each on 1B tokens (the same as 1M parameter proxy models), fitted a regression model, and used the regression model’s predicted data composition to guide the training of a 3B parameter model. As shown in the Table, __the regression model achieves 68.23% in the $\rho$ metric, demonstrating that Go4RL can scale effectively to larger models.__ Therefore, selecting the proxy model size requires balancing computational budget against performance gains, depending on the practical constraints of the computational sources.
>
> |Regression | 60M->60M | 60M->3B|
> | ---: |  :----:   |  :---: |
> |Metric |$\rho$ | $\rho$ |
> | LightGBM | 77.85 | 68.23 |
>
> The experimental results are supplemented in the $\textrm{\color{blue}Appendix D.7}$ of the revision PDF.
>
> ---
>
> > __W3__: Evaluated benchmarks are very out of date.
>
> __A3__: Thank you for your helpful question, which contributes to further improving our approach. The benchmarks MBPP, GSM8K, and Math are adopted by Qwen3`[1]` (published in May 2025), demonstrating that these benchmarks still provide meaningful evaluation of model capabilities. Building on this, we additionally include the following benchmarks: GPQA for general reasoning, LiveCodeBench for code, and LiveMathBench for math. These benchmarks have been used by recently released models such as DeepSeek-V3-exp (Aug 2025)`[2]`, Qwen3 (May 2025)`[1]`, and OLMo-Flex (Sep 2025)`[3]`. As shown in the Table below, the models trained with Go4RL continue to outperform the baselines on these benchmarks, especially in the LiveCodeBench. These results further validate the generality and effectiveness of our method.
>
> | Task|  LiveCodeBench (Code) | LiveMathBench (Math) | GPQA-Diamod (General) | Avg |
> | ---: |  :----:   |  :---: |  :----:   |  :---: |
> |Metric| Acc. | Acc. | Acc.	| Acc. |
> |__RL_loss__ | 3.04 | 0.58 | 23.23 | 8.95|
> |__PPL__| 7.93 | 1.16 | 25.76 | 11.62|
> |__Go4RL_best__ | __14.82__ | 0.69 | __28.79__ | __14.77__ |
>
> > _[1] Yang, An, et al. "Qwen3 technical report." arXiv preprint arXiv:2505.09388 (2025)._
>
> > _[2]  https://github.com/deepseek-ai/DeepSeek-V3.2-Exp/blob/main/DeepSeek_V3_2.pdf_
>
> > _[3] Shi, Weijia, et al. "Flexolmo: Open language models for flexible data use." arXiv preprint arXiv:2507.07024 (2025)._
>
> ---
>
> If there is anything else that requires clarification, we are more than willing to provide further explanation.

---

### Official Review · Reviewer_1jCo · 2025-10-30

**Soundness:** 2
**Presentation:** 2
**Contribution:** 2
**Rating:** 4
**Confidence:** 4

**Summary:**

The authors tried to find the optimal (in a way) pre-training data mix that makes the base LLM perform better on RL tasks. The authors argue that other papers have mentioned that sometimes the RL performance is bottlenecked by the base LLM and they want to find the right data for training the base LLM. The authors proposed a pipeline where the they sample different data mix and train base model, apply RL and create RL models over the base models, compute average perplexity of the base model using the RL model k responses and fit a regression model to find the right data mix given the average perplexity. On downstream tasks, the authors show the gains using the best data mix suggested by the regression model.

**Strengths:**

- The idea is quite interesting to find the best data mix for pre-training that can improve the post-training RL performance. A lot of work has used a base model and then tried to improve the post-training performance by choosing a different data mix and trying with that. The work goes against the norm that pre-training is dead and a lot of effort nowadays is put on the post-training side.
- Experimentation is clear with the benchmarks quite relevant for smaller models. Might be too simple for modern LLMs but quite relevant for smaller models.
- The paper is also clear and easy to understand.

**Weaknesses:**

- The paper compares the pre-training data mix and then RL baselines but I am not sure if simple RL with DPO or GRPO is enough to say that data mix can help in maximizing the RL benefits. What about we don't have to do this at all and comparing with something like Doremi would be beneficial to see if online reweighting is enough to get the same benefits or not.
- Assuming the approach is good even when compared with other data mixing approaches, the scalability of this approach is a big concern. This approach requires training a bunch of proxy models, then RL-fine tuning them to calculate the average ppl and that means there is a substantial cost associated with the approach. Scaling this would be a big challenge.
- Another experimentation weakness related to scaling  I feel is mentioning that regression loss might be better than RL loss or scaling laws but the spearman drops to 58 when the approach is scaled to 1B. And this is just a 1B model. Scaling this further means this approach might not scale at all.
- Finally, I am not sure if I got the justification of the idea right. Minimizing avg ppl based on RL policy distribution encourages the base model distribution to be closed to RL distribution in order to improve the RL performance. I am not sure if that is happening. It might make the model biased in terms of its pre-training data and then that bias is shown in the RL approach too. Getting better outputs in RL samples to get more stable training. This is similar to the cold start problem in RL which can be solved with some warm (usually with some SFT data).

**Questions:**

1. I want to know if there is a link between avg ppl and maximizing RL performance which is not because of the pre-training biases created with the data mix. Does avg ppl correlate with higher GAE ?
2. Can you shed some light on the training cost of all the proxy models and RL models before the right training mix can be found. Can you plot accuracy gains vs the increase in the cost plot?
3. Any comments on why spearman value went down to 58 when scaled to 1B parameters? What does it say about scaling the approach to a much larger model?
4. With the same token / compute budget, can you compare with some post-training data mixing approaches like Doremi ?
5. What's the upper bound of the approach? Is there a way to find the optimal (best) data mix?

---

> ### Author Response · Authors · 2025-11-20
> **Rebuttal by Authors [1/3]**
>
> Thank you for the detailed and insightful discussions on our paper. We hope the following clarifications could provide more clear support for our claims and help address your concerns.
>
> ---
>
> > __W1&Q4__: Compare with something like Doremi would be beneficial to see if online reweighting is enough to get the same benefits or not.
>
> __A1__: Thanks for the insightful suggestion. We conduct experiments to compare with Doremi`[1]`, with the settings as below. We first used the Doremi-released GitHub models to obtain a Doremi-optimized weight of the same pre-training data source (general, code, and math) as our methods, as illustrated in Section 4.1. With the optimized domain weights (math: 0.130, general: 0.723, code: 0.147), we pre-train a 1B parameter base model and then RL-train it to get the RL-trained model. The pre-train and RL-train settings are the same as illustrated in Section 4.1.
>
> The results under GRPO are listed as below. Under the scenario defined in our paper, namely, examining how pre-training dataset mixture strategies relate to obtaining a viable base model for RL training, Go4RL provides a stronger connection between the pre-training and reinforcement learning stages compared with baselines such as Doremi, which focus solely on data selection for pre-training. This enables Go4RL to more effectively identify data mixtures that lead to improved downstream performance after RL.
>
> |Tasks |OBQA | PIQA | Hellaswag | HumanEval pass@10 | MBPP pass@10 | Math | GSM8K | Avg |
> | ---: |  :----:   |  :---: |   :----:   | :---: |   :----:   |  :---: |   :----:   |  :---: |
> |Metric| Acc. | Acc. | Acc. | Acc. | Acc. | Acc. | Acc. | Acc. |
> | __Doremi__ | 12.2 |7.1 | 20.6 | 13.4 | 0.4 | 0.6 |0.2 | 7.8 |
> | __Go4RL_best__ | __12.5__ | __21.2__ |__25.3__ | __36.8__ | __10.5__ | __1.1__ | __0.9__ |__15.5__|
>
> > _[1] https://github.com/sangmichaelxie/doremi_
>
> ---
>
> > __W2&Q2(1)__: This approach requires training a bunch of proxy models, then RL-fine tuning them to calculate the average ppl and that means there is a substantial cost associated with the approach. Scaling this would be a big challenge. The training cost of all the proxy models and RL models before the right training mix can be found.
>
> __A2__: We agree that the computation cost is a big challenge. To further analyze the computation cost, we report the GPU hours and FLOPs required to fit 512 1M-parameter models for predicting the optimal data mixtures of 1B-parameter models, and compare them against other data-selection baselines, listed below.
>
> |Methods	 | PPL | RL_loss |  HUMAN | | Go4RL(ours) | |
> | ---: |  :----:   |  :---: |   :----:   | :---: |   :----:   |  :---: |
> |__Stages__| Pre-training | Total | | Pre-training | RL | Total |
> | __FLOPs__ | 1.95E+19 | 7.45E+19 | 0 | 3.07E+18 | 7.14E+19 | 7.45E+19 |
> |__GPU hours__| 4.08 | 16.14 | 0 | 0.64 |15.5 | 16.14|
>
> As a result, __the cost attributed to training proxy models is minimal—only about 5% of the total compute required to train a 1B base model on 25B tokens.__ The choice of proxy model size and training token budget should therefore be viewed as a trade-off between computational efficiency and final model performance. We've also listed these results in the $\textrm{\color{blue}Appendix D.5}$ of the revised PDF.
>
> ---
>
> > __Q2(2)__: Relationships between accuracy gains vs the increase in the cost?
>
> __A3__: We conducted a study on how proxy-model cost affects accuracy gains. Specifically, we compared the downstream performance of 1B models trained under two proxy configurations: _(1) 300 proxy models with 60M parameters_ and _(2) 512 proxy models with 1M parameters._ All other settings follow those described in the main paper. As shown in the Table, both proxy configurations achieve comparably strong average performance on downstream tasks, while the results of the 60M proxy models do not exhibit a noticeable advantage over the 1M proxy models while incurring substantially higher computational cost, increasing FLOPs from $7.45E+19$ to $4.52E+20$.
>
> |Tasks |OBQA | PIQA | Hellaswag | HumanEval pass@10 | MBPP pass@10 | Math | GSM8K | Avg |
> | ---: |  :----:   |  :---: |   :----:   | :---: |   :----:   |  :---: |   :----:   |  :---: |
> |Metric| Acc. | Acc. | Acc. | Acc. | Acc. | Acc. | Acc. | Acc. |
> | __60M proxy models->1B__| 12.8 | 21.5 | 25.2 | 39.6 | 9.9 | 1.4 | 0.9 | 15.9 |
> |__1M proxy models->1B__ | 12.5 | 21.2 | 25.3 | 36.8 | 10.5 | 1.1 | 0.9 | 15.5|
>
> __According to these results, we recommend prioritizing a larger number of smaller proxy models rather than a few large ones. A suite of small proxy models substantially lowers the barrier to conducting data-mixture studies within a fixed compute budget.__

---

> ### Author Response · Authors · 2025-11-20
> **Rebuttal by Authors [2/3]**
>
> > __W3&Q3__: Scaling this further means this approach might not scale at all. Any comments on why Spearman value went down to 58 when scaled to 1B parameters? What does it say about scaling the approach to a much larger model?
>
> __A4__: Thank you for your valuable question.
>
> To further validate the feasibility of applying Go4RL to larger-scale models, we trained 300 proxy models with 60M parameters each on 1B tokens, fitted a regression model, and used the regression model’s predicted data mixtures to guide the training of a 3B parameter model.
>
> As shown in the Table, the regression model achieves 68.23% in the $\rho$ metric, demonstrating that Go4RL can scale effectively to larger models. When the target model becomes larger, using proxy models with more parameters leads to higher prediction accuracy in the fitted regression model, which aligns with our intuition. Therefore, selecting the proxy model size requires balancing computational budget against performance gains, depending on the practical constraints of the computational sources.
>
> |Regression | 60M->60M | 60M->3B|
> | ---: |  :----:   |  :---: |
> |Metric |$\rho$ | $\rho$ |
> | LightGBM | 77.85 | 68.23 |
>
> The experimental results are supplemented in the $\textrm{\color{blue}Appendix D.7}$ of the revision PDF.
>
> ---
>
> > __W4__: Not sure if I got the justification of the idea right. Minimizing avg ppl based on the RL policy distribution encourages the base model distribution to be close to the RL distribution in order to improve the RL performance. Getting better outputs in RL samples to get more stable training. This is similar to the cold start problem in RL, which can be solved with some warm (usually with some SFT data).
>
> __A5__: We greatly appreciate the time the reviewer has spent on this paper and your understanding is correct. Recent research, such as OctoThinker`[1]` has made a similar observation that "the distribution gap between mid-training data and downstream tasks notably affects RL performance." This could help validate the settings of our method, which could benefit from minimizing the gap in data distribution between the pre-training and RL phases.
>
> With this observation, OctoThinker uses the heuristic method to set up the optimal data mixture during mid-training to turn Llama into a foundation model well-suited for RL scaling. Different from OctoThinker, we leverage a regression-based method to predict the optimal data mixture for pre-training, aiming to minimize the distribution gap between the pre-training and RL stages in a different way. This allows us to obtain a base model that achieves better performance after RL.
>
> > _[1] OctoThinker: Mid-training Incentivizes Reinforcement Learning Scaling, https://arxiv.org/pdf/2506.20512 , June 2025_
>
> ---

---

> ### Author Response · Authors · 2025-11-20
> **Rebuttal by Authors [3/3]**
>
> > __Q1__:I want to know if there is a link between avg ppl and maximizing RL performance which is not because of the pre-training biases created with the data mix. Does avg ppl correlate with higher GAE ?
>
> __A6__: We sincerely appreciate your insightful suggestions, which help strengthen the credibility of our work. We further analyze the correlation between avg-ppl and RL loss with downstream tasks, as well as the correlation between avg-ppl and RL loss themselves.
>
> The higher the correlation between metrics, the larger the absolute value. As shown in the Table, choosing avg-ppl as the target for fitting yields noticeably greater advantages and gains on downstream tasks compared to directly fitting the RL loss ($|−0.57| > |−0.37|$). Additionally, the positive correlation between avg-ppl and RL loss ($value = 0.35$) indicates that avg-ppl and RL loss influence the model in the same direction. These empirical factors reflect the correlation of avg-ppl with the RL performance. With avg-ppl as the fitting target, it could lead to better RL-trained model performance instead of the pre-training biases created with the data mix.
>
> |Tasks| Humaneval | Humaneval_pass@10 | Mbpp | Mbpp_pass@10 | Math | Gsm8k | Obqa | Piqa | Avg. | Rl_loss |
> | :---        |    :----:   |  :---: |    :----:   |  :---: |    :----:   |  :---: |    :----:   |  :---: |    :----:   |  :---: |
> | __Avg(k-ppl)__ (Go4RL) |-0.84  | -0.84 | -0.53 | -0.53 | -0.93 | -0.20 | 0.02 | -0.49 | __-0.57__ | __0.35__ |
> |__Rl_loss__ | -0.49  | -0.50 | -0.18 | -0.18 | -0.17 | 0.69 | -0.15 | -0.52 | __-0.37__ | 1.00 |
>
>
> The updated visual heatmap figure has been added to $\textrm{\color{blue}Appendix D.6}$ of the revised PDF, highlighted in yellow.
>
> ---
>
> > __Q5__: What's the upper bound of the approach? Is there a way to find the optimal (best) data mix?
>
> __A7__: This is a very interesting topic. To the best of our knowledge, existing methods for finding the optimal data mixture are empirical methods such as `[1-2]`. Reference`[3]` is an approximate theoretical method that formally defines and solves the problem of "estimating the impact of training sample orders on model performance without retraining" by approximating the parameter updating process with Taylor expansions. Reference`[4]` presents the first theoretical result on how mid-training shapes post-training. But their targets are not the data mix.
>
> Regarding our method, which relies on a regression-based surrogate trained on a finite set of pilot mixtures, we think its theoretical upper bound is constrained by three factors:
>
> + __Surrogate approximation limit__: The regression model cannot fully reconstruct the true reward landscape, so the method cannot exceed the optimum achievable within the surrogate’s expressiveness.
> + __Finite-sample coverage__: Since the mixture space is continuous but the pilot evaluations are finite, the surrogate is reliable only within the convex hull of sampled mixtures. Therefore, a conservative theoretical upper bound is the best true reward achievable inside this convex hull.
> + __Intrinsic limits of the pretraining–RL pipeline__: Even with the optimal mixture, downstream performance is bounded by model capacity, data quality, and the RL objective. These factors impose a ceiling independent of the mixture optimization.
>
> > _[1] D-CPT Law: Domain-specific Continual Pre-Training Scaling Law for Large Language Models, NeurIPS 2024_
>
> > _[2] Data Mixing Laws: Optimizing Data Mixtures by Predicting Language Modeling Performance, ICLR 2025_
>
> > _[3]Estimating the Effects of Sample Training Orders for Large Language Models without Retraining,https://arxiv.org/abs/2505.22042_
>
> > _[4] Learning to Reason as Action Abstractions with Scalable Mid-Training RL, https://arxiv.org/abs/2509.25810_
>
> ---
>
> We welcome any further inquiries and are glad to provide additional details as needed.

---

### Official Review · Reviewer_JRtj · 2025-10-31

**Soundness:** 3
**Presentation:** 3
**Contribution:** 3
**Rating:** 6
**Confidence:** 4

**Summary:**

This paper proposes Go4RL, a novel framework that predicts optimal data mixture compositions for RL finetuning using a regression based approach. The core idea is to introduce avg(k-ppl) to estimate the reasoning potential of the pretrained model. By fitting regression models to map data mixture ratios to avg(k-ppl), Go4RL predicts high-performing mixtures without costly large scale RL runs. Extensive experiments across different model sizes, domains, and RL algorithms validate the effectiveness of the approach. The results reveal a strong correlation between pretraining data mixture and RL outcomes, suggesting that RL performance is fundamentally constrained by the quality of pretraining data composition.

**Strengths:**

This paper presents a novel and well motivated perspective on understanding the interplay between pretraining data mixture and reinforcement learning (RL) performance in large language models (LLMs). The proposed Go4RL framework and the introduction of the algorithm-agnostic metric avg(k-ppl) represent a clear conceptual innovation that reframes how researchers evaluate the “RL readiness” of base models. The work demonstrates solid methodological quality, supported by extensive experiments across multiple model scales (1M–1B parameters), domains (math, code, and general), and RL paradigms (DPO and GRPO). The experimental evidence is consistent and credible, highlighting the predictive power of regression based data optimization. The manuscript is clearly written, logically structured, and supported by intuitive figures.

**Weaknesses:**

1. The study lacks a systematic analysis of the stability and interpretability of the avg(k-ppl) metric. The paper does not report its variance or robustness under different sampling settings (k values, temperature parameters, generation lengths, or random seeds). If avg(k-ppl) is sensitive to such sampling conditions, its reliability as a regression target for predicting optimal data mixtures could be compromised.
2. The observed degradation on GSM8K is only attributed to data format differences, without further empirical or theoretical analysis. A deeper examination of domain specific interactions would significantly strengthen the interpretive depth of the results. (such as how mathematical versus general text mixtures affect reasoning transfer)

**Questions:**

See Weakness.

---

> ### Author Response · Authors · 2025-11-20
> **Rebuttal by Authors [1/1]**
>
> We first want to thank the reviewer for the thorough review and positive comments. In the rest of this response, we will address the weaknesses and questions raised in the review.
>
> ---
>
> > __W1__: The paper does not report its variance or robustness under different sampling settings. If avg(k-ppl) is sensitive to such sampling conditions, its reliability as a regression target for predicting optimal data mixtures could be compromised.
>
> __A1__: Thanks for the insightful comments. Due to time limitations, we conduct further experiments over some of the different sampling settings, including k values and random seeds. As shown in the Table below, the variation of the settings shows that the $\rho$ values, which indicate regression performance, remain relatively stable. We've also listed these results in the $\textrm{\color{blue}Appendix D.8}$ of the revised PDF.
>
> |    | Methods | $\rho$ (1M->1M)     |
> | ---:        |    :----:   |         :---: |
> | K-values      | K=32     | 87.82   |
> |K-values  | K=128    |86.46 |
> | random_seeds     | random_seeds=1234   | 85.23   |
> | random_seeds     | random_seeds=4321   | 86.19   |
>
> ---
>
> > __W2__: A deeper examination of domain-specific interactions would significantly strengthen the interpretive depth of the results (such as how mathematical versus general text mixtures affect reasoning transfer).
>
> __A2__: Thank you for your valuable suggestions. Based on these, we further visualize the correlation between the ratios of different pre-training data sources and the performance on downstream tasks to analyze how the different ratios of data mixture affect the reasoning transfer.
>
> We use a heatmap to show the relationship between the ratios of pre-training data sources (general, code, and math) and downstream tasks. The heatmap figure has been included in $\textrm{\color{blue}Appendix C.2}$ of the revised PDF, highlighted in yellow. To make it easier to check, we also provide the values for constructing the heatmap figure in Table 1 below (a higher number indicates a greater association between the two variables).
>
> | |OBQA | PIQA | HumanEval | HumanEval pass@10 | MBPP | MBPP pass@10 | Math | GSM8K | Avg |
> | ---:        |  :----:   |  :---: |   :----:   | :---: |   :----:   |  :---: |   :----:   |  :---: |   :----:   |
> | __Math:General__ | 0.31 | 0.45 | 0.53  | 0.60 | 0.20 | 0.45 | __-0.63__ | __-0.55__ | 0.48|
> |__Code:General__| 0.27 | 0.56 | 0.64| 0.87 | 0.39 | 0.87 | __-0.36__ | __-0.36__ | 0.80|
> |__Code:Math__| 0.18 | 0.32 | 0.39 | 0.61 | 0.39 | 0.74 | __0.03__ |__0.07__ | 0.64|
>
>
> It can be clearly observed that increasing the ratios of Math to General, Code to General, and Code to Math data leads to positive effects on most tasks except for the Math-related tasks. In other words, a higher proportion of math-domain pre-training data tends to degrade performance on the GSM8K task. __This observation supports our hypothesis that there exists a noticeable gap between OpenWebMath (which is more oriented toward theorems and proofs) and GSM8K&Math (which focuses more on arithmetic operations such as addition and subtraction) in terms of data format and knowledge coverage.__ We also provided concrete examples from OpenWebMath, GSM8K, and Math in $\textrm{\color{blue}Appendix C.2}$ of the original version, illustrating the differences in data format.  Furthermore, recent research such as OctoThinker [1] has made the similar observation with us.
>
> ---
>
> If there are any additional questions, we would be glad to discuss them.
>
> > _[1] OctoThinker: Mid-training Incentivizes Reinforcement Learning Scaling, https://arxiv.org/pdf/2506.20512 , June 2025_

---

### Official Review · Reviewer_sY5f · 2025-11-06

**Soundness:** 2
**Presentation:** 2
**Contribution:** 3
**Rating:** 4
**Confidence:** 2

**Summary:**

This work introduces a pipeline named Go4RL to select the best mixing ratio of pretraining data for training pre-RL models. It utilizes avg(k-ppl) of the small models trained with different data ratio to train a regression model, which can predict a good ratio for larger 1B model pretraining.

**Strengths:**

1. Try different pretrain data/RL algorithms, train lot of small models and the regression model can predict the result for larger 1B model. Outcoming with human/ppl/rl-loss baselines. In general, I think the experiments look solid
2. The analysis of data mixing ratio makes a lot of sense to me. I also find that mixing more coding data in pre/mid-training is helpful to general reasoning. And recent work also shows that well-RL model (like Qwen) always has some coding epoch during RL training, which may imply some code mid-training.

**Weaknesses:**

1. L51-53 is kind of confusing, I don't get what's the relation between pass@k and average-k ppl. The first one shows the coverage of k-sampling, but the second one does not. Maybe you could compare with pass@k metric on some math/coding/logic tasks, or entropy metric. I think they are also RL-agnostic and wonder if they are better than average-k ppl, because they are more related to downstream metric compared with ppl. (The performance on GSM8k is weaker with avg-k ppl)
2. The writing and figure can be improved. Like Fig. 3 does not have a clear Y axis
3. It seems that when there are new data source, we need to train new regression models, which is very costly. Besides, it's unclear if this method is scalable for larger model, like optimal data mixing ratio of 1M model maybe very different for 32B model, while training a lot of 1B model for regression model is unacceptable. Besides, when number of data source increase, the number for data mixing ratio would increase.

I think the method make sense but may be strongly limited to the generalization problem

**Questions:**

1. What's the parameter of LightGBM / random forest / other baseline used in the paper?
2. What's cost of training regression model? like in gpu hours.

---

> ### Author Response · Authors · 2025-11-21
> **Rebuttal by Authors [1/2]**
>
> Thank you for the detailed and insightful comments on our work. We hope the following clarifications will provide clearer support for our claims and adequately address your concerns.
>
> ---
>
> > __W1__: Maybe you could compare with pass@k metric on some math/coding/logic tasks, or the entropy metric. I think they are also RL-agnostic and wonder if they are better than average-k ppl, because they are more related to the downstream metric compared with ppl.
>
> __A1__:  Thanks for the insightful comments that would help us to strengthen the paper's quality. We firstly explain the relationship between $pass@k$ and average-k ppl. $pass@k$ metrics are used to evaluate the RL-trained model $M_{RL}$'s performance. For each query $q$, $M_{RL}$ samples $k$ responses for $q$, if one of the $k$ responses matches the ground truth, $M_{RL}$ answers correctly for $q$. For the calculation of $avg(k-ppl)$, the above-mentioned $k$ responses are firstly input into the base model, on the basis of which the $M_{RL}$ is trained. Then with the base model, we calculate the average ppl of the $k$ responses and obtain $avg(k-ppl)$. We use $avg(k-ppl)$ instead of $pass@k$ because PPL is more commonly adopted during the pre-training phase of base models as an indicator to measure their performance. Using $avg (k-ppl)$, we could further connect the $M_{RL}$ performance with its base model, achieving a more comprehensive evaluation.
>
> We agree that directly using the $pass@k$ metric or entropy metric could also evaluate the performance. However, we argue that using downstream-task $pass@k$ as the regression target may introduce several inherent limitations:
> + __High Noise and Instability__：$pass@k$ is inherently a discrete and high-variance metric, particularly on code, math, or multi-step reasoning tasks, which injects significant noise into the regression targets and weakens the regressor's ability to learn stable performance trends.
> + __Strong nonlinearity with limited samples__: The relationship between $pass@k$ and the model’s true capability is often highly nonlinear and even piecewise. The regression model may struggle to capture this structure, leading to underestimation of improvements in some regions or overestimation of gains in the high-performance regime.
>
> Our objective is to obtain a metric that can bridge model capability between the pre-training and reinforcement learning stages, which is why we choose to use $avg(k-ppl)$. We further conduct experiments with the same settings to compare the $avg(k-entropy)$ with $avg(k-ppl)$. The experimental results are listed below. According to the results, $avg(k-ppl)$ surpasses the entropy metrics in downstream tasks' performance.
>
> |Tasks | OBQA | PIQA | Hellaswag | HumanEval pass@10 | MBPP pass@10 | Math | GSM8K | Avg |
> | ---:        |  :----:   |  :---: |   :----:   | :---: |   :----:   |  :---: |   :----:   |  :---: |
> | __Metric__ | Acc. | Acc. | Acc. | Acc. | Acc. | Acc. | Acc. |  Acc. |
> | __$avg(k-entropy)$__ | 16.2 |10.6 | 24.9 | 4.3 | 0 | 0.2 | 2.3 |8.4 |
> | __$avg(k-ppl)$ (Go4RL_best)__| 12.5 | __21.2__ | __25.3__ | __36.8__ | __10.5__ | __1.1__ | 0.9 | __15.5__ |
>
> We've also added these results and analysis in the $\textrm{\color{blue}Appendix D.6}$ of the revised PDF.
>
> ---
>
> > __W2__: The writing and figure can be improved. Like Fig. 3 does not have a clear Y-axis
>
> __A2__: We appreciate the valuable comments from the reviewer and have made corrections in the revised version as in the attachments. See $\textrm{\color{blue}Figure 3}$ in the revision PDF for updates, highlighted in yellow.

---

> > ### Author Response · Authors · 2025-11-21
> > **Rebuttal by Authors [2/2]**
> >
> > > __W3__: It seems that when there are new data sources, we need to train new regression models, which is very costly. Besides, it's unclear if this method is scalable for larger models, like the optimal data mixing ratio of 1M model maybe very different for a 32B model, while training a lot of 1B models for a regression model is unacceptable. Besides, when the number of data sources increases, the number of data mixing ratios would increase.
> >
> > __A3__: We agree that， similar to the traditional regression methods, if a new data source comes in and its distribution differs significantly from that of the old data, we need to train new regression models. But as the pre-training data size is usually massive for the base model, e.g., up to 15 trillion tokens, it is difficult for large amounts of new data to become available in a short period of time, which would cause significant changes in the data distribution of the 15T token scale. If the granularity of the data source is finer, the prediction accuracy may improve, but those improvements would lead to a higher frequency of retraining. A balance between cost and performance needs to be achieved. We also list more details of the computation cost in response to your second question below.
> >
> > For the scalable larger models, we further scale up the parameters of the proxy models to 60M and the target model to 3B separately. The pipelines are the same as the 1M proxy models and 1B target model. __The predicted ρ for the 3B-parameter model reaches 68.23%, demonstrating that Go4RL can scale effectively to larger models.__
> >
> > |Regression | 60M->60M | 60M->3B|
> > | ---: |  :----:   |  :---: |
> > |Metric |$\rho$ | $\rho$ |
> > | LightGBM | 77.85 | 68.23 |
> >
> > The experimental results are supplemented in the $\textrm{\color{blue}Appendix D.7}$ of the revision PDF.
> >
> > ---
> >
> > > __Q1__: What's the parameter of LightGBM / random forest / other baseline used in the paper?
> >
> > __A4__: Thanks for the comments, which could help with the reproduction of this work. We list the parameters for the regression methods as below and also in the revised version in the attachment. For the baselines, the settings and implementation details are the same with Go4RL, as illustrated in Section 4.1.
> >
> >
> > |Regression Methods | Hyparameters |
> > | ---: |  :----  |
> > | __LighGBM__ | n_estimators=200 |
> > | | early_stopping_rounds=20 |
> > | | max_depth=3 |
> > | | num_leaves=15 |
> > | | learning_rate=0.05 |
> > | __Ridge Regression__ | alpha=0.5 |
> > | __Random Forest__ | n_estimators=150 |
> > | | max_depth=5 |
> > | | min_samples_split=10 |
> > | | min_samples_leaf=5 |
> >
> > ---
> >
> > > __Q2__: What's cost of training regression model? like in gpu hours.
> >
> > __A5__: We report the GPU hours and FLOPs required to fit 512 1M-parameter models for predicting the optimal data mixtures, and compare them against other baselines. As listed below, the computational cost of our approach remains within an acceptable and marginal range.
> >
> > In addition, our target is to predict the optimal data mixture for larger base models, thereby saving the cost of retraining the base models to get the optimal one. When compared to the entire cost of training a 1B base model on 25B tokens, __training proxy models is rather inexpensive—only roughly 5%__. The effect would be significantly greater when the training data and base model parameters are increased. Therefore, the choice of proxy models should be viewed as a trade-off between computational efficiency and final model performance.
> >
> > |Methods	 | PPL | RL_loss |  HUMAN | | Go4RL(ours) | |
> > | ---: |  :----:   |  :---: |   :----:   | :---: |   :----:   |  :---: |
> > |__Stages__| Pre-training | Total | | Pre-training | RL | Total |
> > | __FLOPs__ | 1.95E+19 | 7.45E+19 | 0 | 3.07E+18 | 7.14E+19 | 7.45E+19 |
> > |GPU hours| 4.08 | 16.14 | 0 | 0.64 |15.5 | 16.14|
> >
> > We've also listed these results in the $\textrm{\color{blue}Appendix D.5}$ of the revised PDF.
> >
> > ---
> >
> > We remain available to address any remaining questions and to offer further clarification whenever needed.

---

### Author Response · Authors · 2025-11-24
**Common responses [1/2]**

We thank all reviewers for their constructive feedback and greatly appreciate all the positive responses. Your comments have been invaluable in further improving both our method and the paper. We have summarized the common concerns raised across the reviews and highlight here several new experimental results that we believe will be of interest to all reviewers.

---

· Is it worth using the proxy models?
---

We report the GPU hours and FLOPs required to fit 512 1M-parameter models for predicting the optimal data mixtures, and compare them against other baselines. $\textrm{\color{blue}As listed below}$, the computational cost of our approach remains within an acceptable and marginal range.

In addition, our target is to predict the optimal data mixture for larger base models, thereby saving the cost of retraining the base models to get the optimal one. When compared to the entire cost of training a 1B base model on 25B tokens, __training proxy models is rather cost-saving—only roughly 5%.__ The effect would be significantly greater when the training data and base model parameters are increased. Therefore, the choice of proxy models should be viewed as a trade-off between computational efficiency and final model performance.

|Methods	 | PPL | RL_loss |  HUMAN | | Go4RL(ours) | |
| ---: |  :----:   |  :---: |   :----:   | :---: |   :----:   |  :---: |
|__Stages__| Pre-training | Total | | Pre-training | RL | Total |
| __FLOPs__ | 1.95E+19 | 7.45E+19 | 0 | 3.07E+18 | 7.14E+19 | 7.45E+19 |
|__GPU hours__| 4.08 | 16.14 | 0 | 0.64 |15.5 | 16.14|

---

· Can Go4RL be extended to a larger model?
---

Regarding whether larger model scales might cause Go4RL to fail, we conduct experiments on models with substantially more parameters to validate the feasibility of applying Go4RL to large-scale models.

Specifically, we trained 300 proxy models with 60M parameters each on 1B tokens, fitted a regression model, and used the regression model’s predicted data mixtures to guide the training of a 3B parameter model. As shown in the $\textrm{\color{blue}Table below}$, __the regression model achieves 68.23% in the $\rho$ metric__, demonstrating that Go4RL can scale effectively to larger models. According to these results, a suite of small proxy models substantially lowers the barrier to conducting data-mixture studies within a fixed compute budget.

|Regression | 60M->60M | 60M->3B|
| ---: |  :----:   |  :---: |
|Metric |$\rho$ | $\rho$ |
| LightGBM | 77.85 | 68.23 |

---

· Why is $Avg(k-ppl)$ superior to other fitting targets?
---

We further analyze the correlation between avg(k-ppl), avg(k-entropy) and RL_loss with downstream tasks. The higher the correlation between metrics, the larger the absolute value. As shown in the $\textrm{\color{blue}Table below}$, choosing avg-ppl as the target for fitting yields noticeably greater advantages and gains on downstream tasks compared to fitting the RL loss and avg(k-entropy) (|−0.57| > |−0.37| > |-0.19|).

|Correlations | Downstream tasks|
| ---: |  :----:   |
|$Avg(k-entropy)$ | -0.19 |
| RL_loss | -0.37 |
| $Avg(k-ppl)$| -0.57|

We further conduct experiments with the same settings to compare the $Avg(k-ppl)$ with other fitting targets. The experimental results are $\textrm{\color{blue} listed below}$. According to the results, $Avg(k-ppl)$ surpasses the other fitting targets in downstream tasks' performance. These empirical factors reflect the correlation of $Avg(k-ppl)$ with the RL performance and with novel $Avg(k-ppl)$ as the fitting target could lead to better RL-trained model performance.

|Tasks | OBQA | PIQA | Hellaswag | HumanEval pass@10 | MBPP pass@10 | Math | GSM8K | Avg |
| ---:        |  :----:   |  :---: |   :----:   | :---: |   :----:   |  :---: |   :----:   |  :---: |
| __Metric__ | Acc. | Acc. | Acc. | Acc. | Acc. | Acc. | Acc. |  Acc. |
| __$Avg(k-entropy)$__ | 16.2 |10.6 | 24.9 | 4.3 | 0 | 0.2 | 2.3 |8.4 |
| RL_loss | 5.8 | 24.5 | 23.0  | 20.1 | 6.0 | 0.9 | 1.9 | 11.7 |
| __$Avg(k-ppl)$ (Go4RL_best)__| 12.5 | __21.2__ | __25.3__ | __36.8__ | __10.5__ | __1.1__ | 0.9 | __15.5__ |

---

> ### Author Response · Authors · 2025-11-26
> **Common responses [2/2]**
>
> In addition, we want to highlight some other new experimental results that may interest all reviewers:
>
> ---
>
> + __Experiments over different sampling settings__: We conduct further experiments on different k values and random seeds, indicating that the regression performance remains relatively stable over different settings.
> + __Relationship between the ratios of pre-training data and downstream tasks__: A higher proportion of math-domain pre-training data tends to degrade performance on the Math tasks, supporting our hypothesis that there exists a noticeable gap between OpenWebMath and GSM8K&Math in terms of data format and knowledge coverage, leading to performance degradation in the math domain.
> + __Compare with strong baseline Doremi__: Go4RL provides a stronger connection between the pre-training and RL stages compared with Doremi, enabling Go4RL to more effectively identify data mixtures that lead to improved downstream performance after RL.
> + __Different Level Proxy Models__: We use 60M models as proxy models to generate optimized data mixtures for training 1B models, showing that prioritizing a larger number of smaller proxy models is good enough rather than applying a few large ones.
> + __More benchmarks to evaluate__: We additionally include the following benchmarks: GPQA for general reasoning, LiveCodeBench for code, and LiveMathBench for math. The models trained with Go4RL continue to outperform the baselines on these benchmarks, further validating the generality and effectiveness of our method.

---

### Author Response · Authors · 2025-11-27
**Revised Paper**

In general, we express our gratitude to the reviewers for their invaluable feedback, and have revised and re-uploaded the paper based on the reviewers' suggestions. The main changes are noted in yellow. The updated part primarily includes:

---

○ $\textrm{\color{blue}Figure 3}$: add the Y axis to make the figure clearer.

○ $\textrm{\color{blue}Appendix C.2}$: use a heatmap to display the relationship between the ratios of pre-training data sources ○ (general, code, and math) and downstream tasks under GRPO.

○ $\textrm{\color{blue}Appendix D.5}$: report the GPU hours and FLOPs required to fit 512 1M-parameter models for predicting the optimal data mixtures.

○ $\textrm{\color{blue}Appendix D.6}$: analyze the correlation between avg(k-ppl), RL loss and avg(k-entropy) with downstream tasks.

○ $\textrm{\color{blue}Appendix D.7}$: extend Go4RL to a larger 3B model.

○ $\textrm{\color{blue}Appendix D.8}$: conduct further experiments over different sampling settings.

---

### Note · Authors · 2026-03-27

I have read and agree with the venue's withdrawal policy on behalf of myself and my co-authors.

---

### Meta-Review · Area_Chair_qxCa · 2026-01-07

**Summary:**

RL-trained large language models have shown strong reasoning abilities, but their performance is often limited by the capabilities of their base models. This paper introduces Go4RL, a framework for identifying base models that are better suited for reinforcement learning. Go4RL measures a base model’s reasoning boundary using its perplexity on RL-generated responses and frames data-mixture selection as a regression problem guided by this metric. By predicting and selecting optimal pre-training mixtures, Go4RL produces base models that consistently lead to stronger RL performance.

The reviewers' concerns include that the method: (1) cannot be generalized to new pretraining data source, (2) is not scalable and has high computational expense, (3) depends on proxy models with uncertain scalability to large-scale training, and (4) is evaluated only on outdated benchmarks and lack of detailed analysis.

**Reviewer Concerns:**

Reviewer sY5f's concern is the method cannot be generalized to the new pretraininig data source. I don't think it is addressed during rebuttal.

Reviewer JRtj's concern is lack of detailed analysis in the experiments. I think this is addressed during rebuttal.

Reviewer 1jCo's concern is the scalability of the proposed approach, which requires training a bunch of proxy models. I think it is partially addressed.

Reviewer d5ci's concern is the proposed method (1) is computationally expensive, (2) depends on proxy models with uncertain scalability to large-scale training, and (3) is evaluated only on outdated benchmarks. (3) was addressed during rebuttal. (1) and (2) are not addressed.

**Reviewer Scores:**

Reviewer sY5f's score is 4, and I don't think they will increase the score.

Reviewer JRtj's score is 6, and I don't think they will increase the score.

Reviewer 1jCo's score is 4, and I think they may increase the score.

Reviewer d5ci's score is 2, I think they might increase the score to 4, but not 6 or above.

---

### Decision · Program_Chairs · 2026-01-26

Reject